# Large-scale electrical resistivity tomography in the Cheb Basin (Eger Rift) at an ICDP monitoring drill site to image fluid-related structures

Tobias Nickschick[1], Christina Flechsig[1], Jan Mrlina[3], Frank Oppermann[2], Felix Löbig[1], and Thomas Günther[2]

[1]Institute for Geophysics and Geology, Leipzig University, Talstrasse 35, 04103 Leipzig, Germany
[2]Leibniz Institute for Applied Geophysics, Stilleweg 2, 30655 Hannover, Germany
[3]Institute of Geophysics CAS, Boční II 1401, 141 31 Praha 4, Czech Republic

**Correspondence:** Tobias Nickschick (tobias.nickschick@uni-leipzig.de)

**Abstract.**

The Cheb Basin, a region of ongoing swarm earthquake activity in the western Czech Republic, is characterized by intense carbon dioxide degassing along two known fault zones - the N-S-striking Počatky-Plesná fault zone (PPZ) and the NW-SE-striking Mariánské Lázně fault zone (MLF). The fluid pathways for the ascending $CO_2$ of mantle origin are subject of the International Continental Scientific Drilling Program (ICDP) project "Drilling the Eger Rift" in which several geophysical surveys are currently carried out in this area to image the topmost hundreds of meters to assess structural situation, as existing boreholes are not sufficiently deep to characterize it.

As electrical resistivity is a sensitive parameter to the presence of conductive rock fractions as liquid fluids, clay minerals and also metallic components, a large-scale dipole-dipole experiment using a special type of electric resistivity tomography (ERT) was carried out in June 2017 in order to image fluid-relevant structures. We used permanently placed data loggers for voltage measurements in conjunction with a moving high-power current sources for generating sufficiently strong signals that could be detected all along the 6.5 km long profile with 100 m and 150 m dipole spacings. After extensive processing of time series for voltage and current using a selective stacking approach, the pseudosection is inverted which results in a resistivity model that allows reliable interpretations depths of up than 1000 m.

The subsurface resistivity image reveals the deposition and transition of the overlying Neogene Vildštejn and Cypris formations, but also shows a very conductive basement of phyllites and granites that can be attributed to high salinity or rock alteration by these fluids in the tectonically stressed basement. Distinct, narrow pathways for $CO_2$ ascent are not observed with this kind of setup which hints at wide degassing structures over several kilometers within the crust instead. We also observed gravity/GPS data along this profile in order to constrain ERT results. A gravity anomaly of ca. -9 mGal marks the deepest part of the Cheb Basin where the ERT profile indicates a large accumulation of conductive rocks, indicating a very deep weathering or alteration of the phyllitic basement due to the ascent of magmatic fluids such as $CO_2$. We propose a conceptual model in which certain lithologic layers act as caps for the ascending fluids, based on stratigraphic records and our results from this experiment, providing a basis for future drillings in the area aimed at studying and monitoring fluids.

# 1 Introduction

Over the last decades, methods that study the electric resistivity of the subsurface - such as magnetotellurics (e.g. Muñoz et al. 2018 or Blecha et al. 2018) and Electrical Resistivity Tomography (e.g. Storz et al. 2000, Schütze and Flechsig 2002, Schmidt-Hattenberger et al. 2013) - have proven to be especially useful when fluids are involved, as they are used as efficient techniques for non-invasive imaging of subsurface structures. A multitude of experiments that focus on carbon dioxide in particular have been carried out, mainly at carbon capture and storage sites, which are typically well explored and where the fluid injection system is controllable (Carrigan et al., 2013; Nakatsuka et al., 2010; Schmidt-Hattenberger et al., 2013; Bergmann et al., 2017). However, when it comes to natural $CO_2$ emanation sites, such as volcanically or magmatically active areas, this is often not the case: the fluid system can often be very complex and variable in space and time and hence requires special approaches (Finizola et al., 2009; Pettinelli et al., 2010; Revil et al., 2008, 2011; Gresse et al., 2017). Furthermore, many methods that focus on fluids are often limited in their resolution and/or depth when it comes to studying those fluids, their migration and interactions with the host rock. One major target site for this kind of studies is the western Eger Rift in Central Europe that has been a center of research for various mantle gas and fluid-related studies within the last 2 decades. It can be called a natural analogue to carbon capture and storage sites where methods used for the detection and monitoring of $CO_2$ and fluids in general can be applied to great success (Schütze et al., 2012).

Particularly the Cheb Basin, located in W-Bohemia/CZ near the border between Germany and Czech Republic, represents the western part of the Eger Rift - the easternmost segment of the European Cenozoic Rift System (Fig.1, Ziegler 1992; Ziegler and Dezes 2007). The area is characterized by ongoing magmatic processes in the intra-continental lithospheric mantle. The most recent article on that topic, Hrubcová et al. (2017), hypothesize that this is caused by magmatic underplating. These processes take place in absence of any currently active volcanism at the surface - the latest activity known is linked to the eruption of two scoria cones (Železná hůrka and Komorní hůrka) and two maar-diatreme volcanoes (Mýtina maar and Neualbenreuth maar, Mrlina et al. 2007, 2009; Flechsig et al. 2015; Rohrmüller et al. 2018). However, they are expressed by a series of geodynamic phenomena like the occurrence of repeated earthquake swarms, surface exhalations of mantle-derived and $CO_2$-enriched fluids in mofettes and mineral springs, and neotectonic crustal movements, which are not expected to occur in an intra-plate regions (Bräuer et al., 2008, 2009; Fischer et al., 2014).

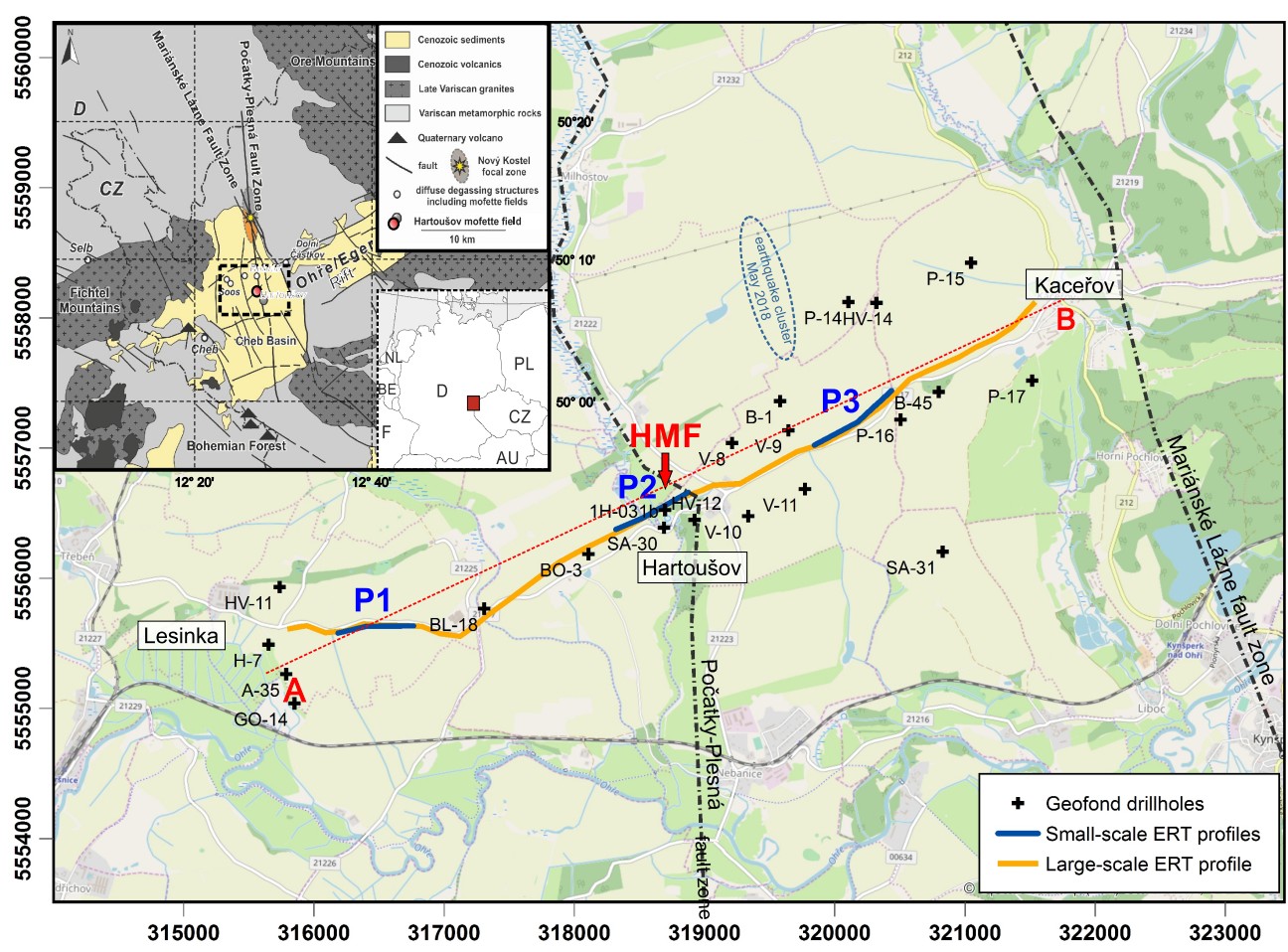

**Figure 1.** Map of the measured large-scale ERT profile (6.5 km), small-scale 625-700 m long ERT profiles (P1, P2, P3), and existing drill holes (Czech Geological Survey) with lithological information. All coordinates are in UTM zone 33N. Red, dotted line marks the location of the lithological transect in Fig. 2. The Počatky-Plesná zone (PPZ) and Mariánské Lázně fault zone (MLF) are drawn as the main tectonic features. HMF = Hartoušov mofette field. Inset: Geological sketch map of the western Bohemia/Vogtland area and the Cheb Basin near the German-Czech border in Central Europe, modified from Flechsig et al. (2008); Dahm et al. (2013); Bussert et al. (2017).

At present, the highest release of energy via earthquakes since 1985 and the emission of mantle-derived $CO_2$ takes place in the Cheb Basin - the former in the area around Nový Kostel, the latter at the Bublák and Hartoušov mofette fields at the surface, which is approximately 10 km south of the Nový Kostel focal area (Fig. 1). Earthquake swarms are sequences of hundreds or thousands of earthquakes with low to moderate magnitudes, mainly without a main- and aftershock behavior which occur over weeks or months and which are typical for recent active volcanic, hydrothermal or geothermal regions. Fluids are involved in these sequences, but their propagation and dissipation within the earth's crust has not yet been fully clarified. Several authors have discussed the potential influence of these fluids in triggering the earthquake swarms, in which the $CO_2$-dominated fluids

of mantle origin migrate through the lithosphere and how they are expected to act on fault zones (Weinlich et al., 1998; Heinicke and Koch, 2000; Bräuer et al., 2005, 2008, 2009; Kämpf et al., 2013; Fischer et al., 2014; Hainzl et al., 2016), but the relation between earthquake swarms and $CO_2$ degassing is still in discussion (e.g. Babuška et al., 2016). The main focus of the current International Continental scientific Drilling Program (ICDP) project "Drilling the Eger Rift" is to understand

the processes behind the origin of the swarm earthquakes in relation to the fluid and $CO_2$ ascent, and their movement through and within the subsurface ("fluid triggered lithospheric activity") supported by a network of five boreholes (maximum depth 400 m) which serve different seismological, microbiological and fluid monitoring aspects (Dahm et al., 2013). One of these key drill sites, the Hartoušov mofette field (HMF) near the village of Hartoušov, will consist of three separate drill holes of different depths (30, 108 and approximately 400 m) which will serve as monitoring stations for gas signature analyses,

innovative sampling/monitoring of fluids and microorganisms, and seismological measurements. This drilling site was selected according to preliminary geological and geophysical investigations conducted in the area of the mofette field (Flechsig et al., 2008; Kämpf et al., 2013; Sauer et al., 2013; Schütze et al., 2012; Nickschick et al., 2015; Bussert et al., 2017) with information about the first 80-100 m.

    Within the ICDP project "Drilling the Eger Rift", we carried out a field experiment using large-scale electrical resistivity

tomography (ERT, Fig. 1) as the favorable geophysical method to detect fluid signatures within the geological units to provide information about their migration through the basin, based on electric resistivity. The method was chosen due to its high sensitivity to pore properties (porosity, salinity, fluid/gas content), as well as clay content. Profile lengths of more than 6 km are necessary to obtain investigation depths of over 1000 m and to resolve structures at this depth sufficiently precisely. ERT has proven to be a useful exploration technology for many geological, environmental and engineering survey problems, since

computerized multi-electrode devices composed of transmitter and receiver in one unit are available. Unfortunately, the use of multi-electrode devices is limited to small layouts (approximately 100 electrodes and spacing of 5-20 m in most cases between the electrodes), resulting in near surface investigation depths of several tens of meters. In order to gain insight into greater depths, specific investigation strategies (dipole-dipole arrays), equipment (high power sources and separate data loggers for voltage measurements) and extensive data processing are necessary.

First theoretical considerations and practical tests for deep electrical sounding with dipole-dipole arrays are documented by Alfano (1974) and Alfano et al. (1982). Because of the logistical effort of large-scale ERT, just a few experiments with exploration lines up to approximately 20 km are documented. Storz et al. (2000) imaged geological units and fault zones at the German continental deep-drilling site KTB ("Kontinentale Tiefbohrung") on a profile up to 20 km. Schütze and Flechsig (2002) conducted a 22 km profile across the Long Valley caldera volcano. The results reveal prominent conductivity structures

interpreted as faults with circulating hot fluids and the present-day flow regime of hydrothermal fluids (Pribnow et al., 2003). Günther et al. (2011) described how a fault zone can be imaged with large-scale ERT and additional structural information from seismics along a 2.5 km long profile. Bergmann et al. (2017) used a surface-downhole ERT survey line (approximately 4-5 km) for monitoring the progress of carbon dioxide sequestration at Ketzin, Germany. Ronczka et al. (2015) used iron boreholes as long electrodes to investigate inland saltwater intrusion into a 4x4 km wide area. Flechsig et al. (2010) conducted

a feasibility survey in a 20x20 km area inside the Eger rift zone as a first test for this method's suitability in this particular

area with industrial noise. A coarse block model was derived from the sparsely distributed current and voltage dipoles and the incorporation of known geological and structural information, such as faults and lithological units. It could be demonstrated that even under noisy conditions, artificial signals can be measured over distances of more than 10 km with sufficient quality despite the electrical noise sources in the Eger Rift, such as power lines, power plants, or from machines used in lignite mining.

Our study specifically focuses on the main fluid escapement center - the Hartoušov mofette field. This particular site is characterized by sediment coverage of ≈85 m, shows high and widely distributed $CO_2$ flux (Kämpf et al., 2013; Nickschick et al., 2015), a phyllitic basement and is situated at a known N-S striking fault zone (Počatky-Plesná fault zone – PPZ, after Bankwitz et al. 2003b). The SW-NE trending ERT profile presented here, measured in June of 2017 features a total length of about 6.5 km and crossed the proposed ICDP drill site and the surface traces of the PPZ. Additional results from several ERT

profiles with lengths of 100-750 m and an investigation depth of about 80 m are available and had been partly conducted before and during the survey campaign (Flechsig et al., 2010; Nickschick et al., 2015).

The key aspects of the geoelectrical research and expected contributions to answer the following scientific aims are:

1. to image the electrical resistivity distribution and characteristics in a near surface scale of approximately 1000 m including the interpretation of the structural patterns: Which characteristic geological and structural settings and geometries of

the resistivity distribution in the subsurface of the target areas with a resolution less than 50 m are evident? What is the lateral/spatial extension of the fault zone derived from the resistivity distribution?

2. to image the possible fluid pathways and the feeding system of the degassing area: Which structures are linked to the migration of $CO_2$? Do we recognize potential structures acting as a fluid trap?

3. to identify characteristic tectonic structures caused by the ongoing geodynamic processes. Is it possible to find weakness

zones which can act as permeable fluid transport pathways?

4. to establish a reference resistivity subsurface model for possible future long term monitoring projects.

## 2   Survey area

### 2.1   Geology and geodynamic activity

The Cenozoic Eger Rift with the central Eger Graben, the NNW-SSE trending Mariánské Lázně fault zone (MLF), and the

Cheb-Domažlice Graben are prominent tectonic structures of the Bohemian Massif, which is the eastern part of the European Cenozoic Rift System (Bankwitz et al., 2003b; Malkovský, 1987; Ziegler, 1992; Peterek et al., 2011). The Eger Rift contains several basins (e.g. Cheb Basin, Sokolov Basin, Most Basin) with similar sedimentary and tectonic evolution (Pešek et al., 2014). The investigation area, the geodynamically active Cheb Basin, a shallow Neogene intra-continental basin with maximal depth of approximately 350 m, was formed at the intersection of the NE- SW striking Eger Graben and the NNW-striking Cheb-

Domažlice Graben (Špičáková et al., 2000; Peterek et al., 2011). The Cheb Basin is bounded on its eastern side by the morphologically distinct scarp of the NNW-SSE trending Mariánské Lázně Fault, and the down dipping Smrčiny/Fichtelgebirge

Mountains to the west and the Bohemian Forest to the south (Fig. 1, Peterek et al. 2011; Bussert et al. 2017). At the west and east border of the Cheb Basin, the basement has an offset of more than 200-400 m. To the north and south, the bottom of the basin thins out gradually to the surface (Bankwitz et al., 2003b; Rojik et al., 2014).

Babuška et al. (2007) point out that the Cheb Basin is located above a triple junction of the Variscan crustal units of the Saxothuringian in the Northwest, the Teplá-Barrandian in the central region, and the Moldanubian in the Southeast. The basin is embedded into Proterozoic and Paleozoic magmatic and metamorphic rocks of the north-western Bohemian Massif - predominantly granites, gneisses, mica schists and phyllites. The sedimentary fill of the Cheb Basin around the area of interest itself consists mainly of less than 300 m of continental clastics (representing debris of these rocks (Bussert et al., 2017), Fig. 1) and overlies the deeply weathered mica schists with interbeds of metaquartzite, metabasite and crystalline limestone which are intruded by granitoid plutons (Variscan Smrčiny, Fichtel and Žandov plutons, Pešek et al. 2014). Several uplift and subsidence events due to varying extensional and compactional stress within the Eger Rift since the Eocene affected the sedimentation within the basin (Peterek et al., 2011; Pešek et al., 2014; Rojik et al., 2014; Bussert et al., 2017). After local deposition of clays and sands in the Eocene (Staré Sedlo formation), sedimentation continued with the deposition of Oligocene to Miocene gravel, sand and clays (named Lower Argillaceous-Sandy formation or Lower Clay-Sand formation). During the Lower Miocene, wetlands dominated the area and let to the deposition of the coal- and lignite-bearing Main Seam formation. As the result of ongoing tectonic activity, a lake developed in which the clay-dominated Cypris formation was deposited. After a hiatus, sedimentation started again in the Pliocene with lacustrine clays, sands and gravels of the Vildštejn formation and continued without an obvious break into the Quaternary.

Currently, the area around Nový Kostel (Fig. 1) is the most active earthquake swarm zone in W-Bohemia/Vogtland (Fischer et al., 2014). The activity at the Nový Kostel focal zone is supposed to be related to the re-activation of a system of faults, e.g. at the intersection between the NNW-SSE trending MLF and the N-S trending PPZ. The earthquake foci are located at depths between 6 and 13 km, clustered along vertical faults, forming an almost continuous, about 15 km long belt striking NNW to SSE and steeply dipping westwards (Fischer and Michálek, 2008; Fischer et al., 2014). Normal and strike slip faulting are the typical focal mechanisms for these intraplate events here. Most of the micro-earthquakes hypocenters are aligned in a N-S direction and thus follow the course of the PPZ, whereas the NNW-SSE striking MLF seems to be only partially seismically active (Bankwitz et al., 2003b; Fischer et al., 2014). The PPZ forms an escarpment of more than 20 m height in Pliocene/Pleistocene sediments and has probably been active since the late Pleistocene (Bankwitz et al., 2003b; Peterek et al., 2011; Bussert et al., 2017). Strike-slip faults with a vertical component run across the basin in E-W direction (e.g., Nová Ves fault) according to Bankwitz et al. (2003a). The combination of seismological and especially hydrological analyses points out that the Nový Kostel zone is also part of the gas uplift system and must be linked to the near surface water flux. The model, which Neunhöfer and Hemmann (2005) proposed, provides an explanation of the active ascent of fluids on the phenomenon of earthquake swarms. The model takes a special two-phase system formed by water and $CO_2$ in contrast to other mixed models (Bräuer et al., 2008, 2009) into account. Furthermore, Horálek and Fischer (2008) assumed that ascending crustal fluids could play a key role in the alteration of the pre-existing, favorably oriented faults from subcritical to critical state due to pore pressure increase. Although ascending fluids from deep crustal root zones are considered as the main reason for inducing recurring earthquake swarms by

pore pressure increase (Špičak and Hóralek, 2001; Weinlich et al., 1998; Heinicke and Koch, 2000; Weise et al., 2001; Bräuer et al., 2005, 2009; Kämpf et al., 2013; Fischer et al., 2014), the relation between earthquake swarms and the source of $CO_2$, $CO_2$ ascent and degassing is still a matter of discussion (Babuška et al., 2016).

One of the main fluid discharge centers for carbon dioxide via mofettes at the surface are located approx. 10 km south of Nový Kostel along the course of the PPZ (Bublák and Hartoušov mofette fields). Only isolated $CO_2$ vents and mineral springs are found close to the MLF (e.g. Dolni Častkov mofette). The numerous cold $CO_2$ emanations with >99 vol % $CO_2$ and mantle signature (He and N isotopes) are supposed to be generally connected to the seismic activity and to stem from upper mantle reservoirs (Weinlich et al., 1998; Geissler et al., 2005; Bräuer et al., 2009, 2011). From the high gas flux rates and high $^3He/^4He$ ratios, the mofette field Bublák-Hartoušov appears to act as deep-seated fluid migration zone along the PPZ (Bräuer et al., 2011; Kämpf et al., 2013). The tectonic setting of the area is of great influence on the increased degassing of $CO_2$ at the surface. Since the early work of Irwin and Barnes (1980), it has become evident that a close relationship exists between the tectonic activity and anomalous crustal emissions of $CO_2$. Due to their hydraulic permeability, faults can act as preferential pathways for the upward migration and release of deep fluids to the atmosphere in this area (Bankwitz et al., 2003a; Geissler et al., 2005). At surface, $CO_2$ emission occurs often at gas vents with diameters <1 m (Kämpf et al., 2013; Nickschick et al., 2017) with high flux rates, and in moderate amounts diffusely over the larger area in general (Kämpf et al., 2013; Nickschick et al., 2015, 2017, see also section 2.2). However, the deep structure, geometry, and lateral extension due to the depth of the fluid pathways in the crust layers are still unknown. Despite the geodynamic-geophysical, and especially seismological research (Švancara et al., 2000; Růžek and Horálek, 2013; Fischer et al., 2014) in this area, many questions about the settings for the fluid regime and the generation of the earthquake swarms remain unanswered. Besides the local and regional stresses, as well as contrasts in rheological rock properties, the fluid Movement and distribution is an essential factor influencing the seismicity of the region. One peculiar phenomenon is the spatial separation of the earthquakes near Nový Kostel and the $CO_2$ degassing near Hartoušov, despite having a similar source behind them. However, in May 2018, a cluster of several (>70) small-magnitude earthquakes was registered (Czech PEPIN seismological catalogue, www.ig.cas.cz) a few hundreds of meters to the NE of the mofette field Hartoušov.

## 2.2 Existing geophysical results and lithological data

From previous geoelectrical investigations, results from several 2D ERT profiles with lengths of 100-750 m, and an investigation depth of approx. 80-100 m across the main faults of the Cheb Basin (MLF and PPZ, Fig.1 are available (Flechsig et al., 2008, 2010; Fischer et al., 2014; Nickschick et al., 2015, 2017; Blecha et al., 2018). The obtained resistivity models reveal the characteristics and width of the fault zones in the shallow subsurface by means of resistivity anomalies, variations in sediment thickness and vertical layer displacement. Significant resistivity anomalies in the subsurface reveal the location of both MLF and PPZ and typical conductive features indicate potential fluid transport paths and regions with mineral alteration. Essentially, both fault zones are characterized by an extended subsurface region (100-250 m) controlled by multiple, more or less parallel, sub-faults with different strike angles. As a local comparative geoelectric (3D small scale ERT), soil gas and sediment study of a $CO_2$ degassing vent in the Hartoušov mofette field, near surface structures to a depth of 20 m were investigated by Flechsig

et al. (2008). The investigations reveal substantial structural features that are to be directly or indirectly related to high $CO_2$ flow (anomalies of electrical resistivity, self-potential, and sediment properties). With the aim to reach deeper structures up to 5 km, several magnetotelluric investigations in the western margin of the Bohemian Massif and along the 9HR seismic profile (Cerv et al., 1997, 2001; Pícha and Hudeková, 1997; Di Mauro et al., 1999) have been carried out since the 1990 resulting in very coarse conductivity models.

Recent information about the regional distribution of electrical resistivity up to 25 km depth came from a 2D magnetotelluric experiment on a 50 km long N-S profile with 25 stations crossing the Cheb Basin in 2017 (Muñoz et al., 2018). The most prominent deep reaching structure is a channel of higher conductivity compared to the surrounding, which extends from the surface at the mofette field of Bublák-Hartoušov into the lower crust (approximately 25 km) to the north, possibly correlated with the hypocenters of the seismic events of the Nový Kostel focal zone. This channel has been interpreted by the authors as a pathway from a mid-crustal fluid reservoir to the surface along deep reaching faults. Whereas the overall resistivity is very high (> 500 - 1000 Ωm) in great parts of the model, very low resistivity (<30 Ωm) could be found near the surface at the mofette fields of Bublák and Hartoušov and their feeding system. Further relevant data and information from other geophysical methods for interpretation of the measured ERT profile are not available or not in the necessary scale.

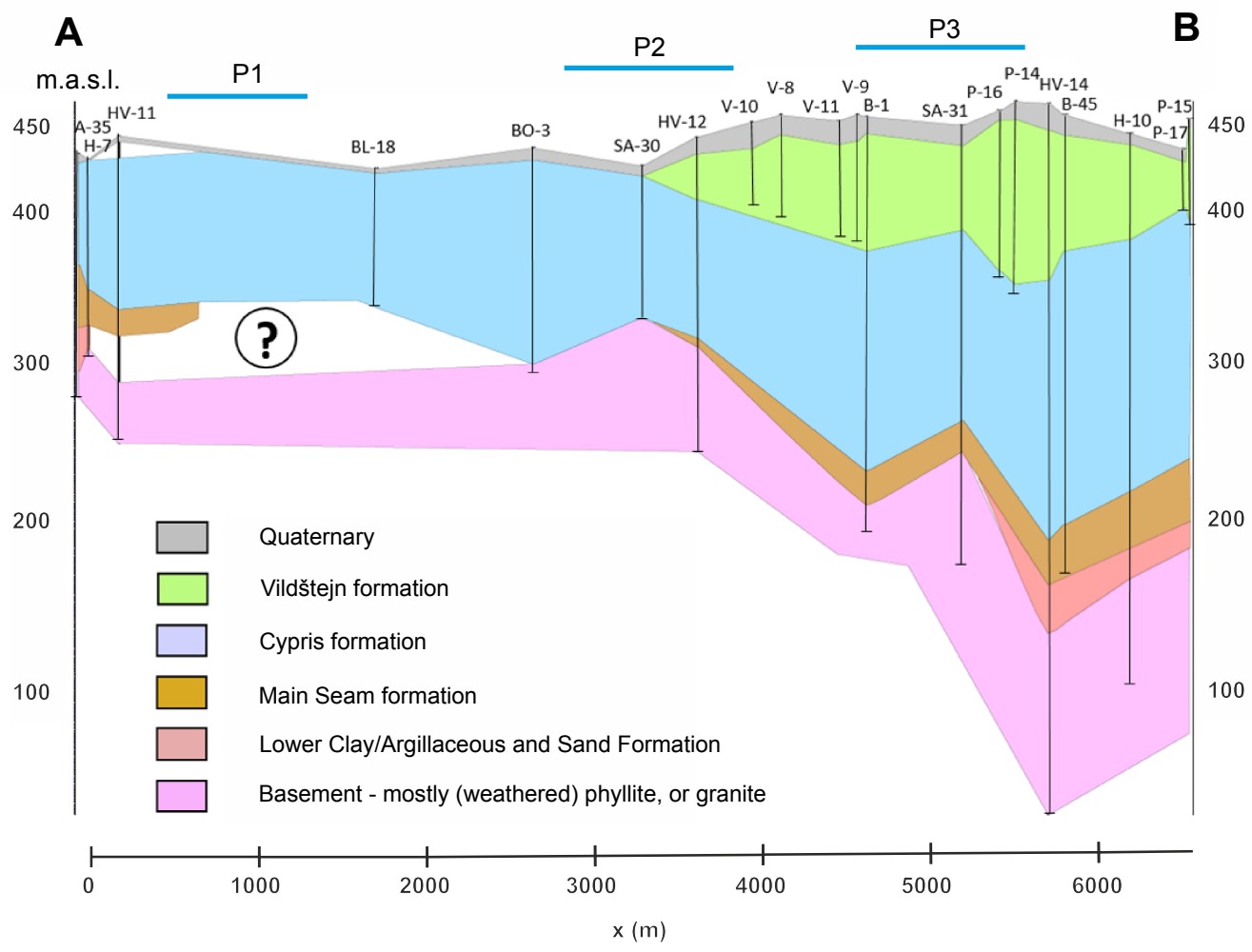

**Figure 2.** Lithological transect along the large-scale profile, based on the descriptions of boreholes from the Czech Geological Survey (former GEOFOND). Question mark indicates an area of unknown lithology and the uncertainty of whether Main Seam and Lower Clay (or Argillaceous-)-Sand formation are present in this area. P1 - P3 mark the locations of the small-scale ERT profiles. For each drill's location, please refer to Fig. 1.

To interpret the subsurface resistivity situation around our survey's target, borehole descriptions from the Czech Geological Survey (former GEOFOND) were gathered. In order to establish a conception of the encountered lithologic units in this experiment, we generated a 2D transect based on the borehole data to a depth of 50 to 400 m (Figs,1 and 2). From the available drills in the investigation area, we selected 20 that provided sufficient depth and were closest to our ERT profile. The GeODin software was used to generate the transect that can be seen in Figure 2. Please note that none of these drills have reached the crystalline phyllite in its unweathered state and only describe the basement phyllite as weathered or highly weathered. In addi-

tion to this geological constraint, we regarded the results from Dobeš et al. (1986): Their report contains valuable petrophysical information from previous studies about the different stratigraphic units in and below the Cheb Basin which we have summarized in Tab. 1. The phyllitic-granitic basement is characterized by low porosities of less than 5% compared to the sedimentary deposits on top, which feature porosities of 15-30%. Resistivity, however, may vary drastically, depending on heterogeneities

within the sediments and whether fluids such as mineral waters or $CO_2$ are present or not and the report does not specifically state where the samples were taken from. For this area, Bussert et al. (2017) provides additional information. Not only do they mention the occurrence of highly mineralized water in the central part of the HMF, their geophysical log of the HJB-1 drill reveals resistivities of 5-10 $\Omega$m for the Cypris formation and 10-20 $\Omega$m for the topmost part of the weathered phyllites. They are about one order of magnitude lower than the values presented in Dobeš et al. (1986) - stressing the importance of regarding

the occurrence or absence of fluids even more.

**Table 1.** Petrological description of the stratigraphic layers of sediments in the Cheb Basin and the basement below, translated from Dobeš et al. (1986).

| Name of stratigraphic unit | Rock type | Porosity [%] | electrical resistivity [$\Omega$m] | |
|---|---|---|---|---|
| | | | minimum-maximum | average |
| Vildštein | gravel, sand, clay | 30.0 | 14-1600 | 350 |
| Cypris | clay, silt, carbonates | 14.5-21.5 | 50-1500 | - |
| Main Seam | lignite, sand, clay | 22 | 7-50 | 15 |
| Lower Sand & Argillaceous | gravel, sandy clay | - | 3-150 (depending on saturation) | 7.5 |
| Phylliitic basement | weathered phyllite | 3.2 | 75-140 | 110 |
| Phyllite basement | unweathered phyllite | 1.0 | 500-1800 | 890 |
| Granitic basement | granite | 5.0 | 65-650 (weathered); > 650 for unweathered | - |

## 3   Methodology

The resistivity of rocks is notably sensitive to the presence of fluids that dominate the conductivity over the rock matrix, and weakening effects of the rock matrix due to fluid-rock interactions. Therefore, ERT is qualified for the detection of fluid signatures in the subsurface structures in different scales, like fluid pathways and fluid-rock interactions processes. Modern

ERT inversion and modeling techniques (Günther, 2004; Günther et al., 2006) can then been applied to the data to retrieve a conductivity image in detail. In the frame of this experiment, one large-scale profile and several small-scale profiles were carried out in June 2017. The SW-NE trending 6.5 km profile crossed the proposed ICDP drill site (Dahm et al., 2013; Bussert et al., 2017) at the HMF and the surface traces of the N-S trending PPZ. Figure 1 shows a location map with existing boreholes and the individual ERT profiles that are discussed subsequently.

## 3.1 Large-scale ERT survey

The data acquisition was performed using the dipole-dipole configuration (AB MN, with A and B being the current injection electrodes and M and N being the potential electrodes) which is, considering the cost-effect-relation for practical and theoretical reasons, most suitable for this large-scale ERT experiment. Transmitter and receiver units are physically separated on two lines reaching maximum dipole separations of 6.5 km (Fig. 1) while keeping the total length of required cables to a minimum as only neighbouring electrodes have to be connected. Considering crop growth in June in this rural area and traffic by agricultural farming machines in general, other arrays are not effective with large cable spreads of several kilometres. Furthermore, we expected vertically oriented features (faults, "fluid channels"), as seen in previous studies (Nickschick et al., 2015), supporting the choice of using a dipole-dipole setup and achieving good results in previous studies at different location with a similar setup (Flechsig et al., 2010; Pribnow et al., 2003; Schmidt-Hattenberger et al., 2013).

The experiment setup included 59 transmitter and voltage dipole locations by using 150 m dipole lengths in the outer (10 dipoles in the western and 11 in the eastern part of the profile) and 100 m length in the central part. While the receivers are stationary at fixed places during the campaign, the transmitter with the source dipole is moved to the feeding positions. Since the profile crosses streets and rural roads, small gaps needed to be left out for current injections and voltage registrations, leading to a total number of 54 voltage reading positions and 47 current injections. To determine the horizontal position, we used a handheld GPS (Garmin GPS map 62s) with an accuracy of about 3 m. Elevations were then taken from a high-resolution digital elevation model. Two high power transmitter (10 kW SCINTREX TSQ-4 and a self-developed 40 kW power transmitter) were used to inject a square-wave signal with a 8 seconds signal period and 50% duty cycle (2 s positive, 2 s off, 2 s negative, 2 s off) and using at least six cross-shaped, stainless-steel metal rods (1.5 m long) for grounding. For a total length of 20 minutes, current was injected. For 15 of these 20 minutes (112 total periods), we injected with the highest current possible, resulting in clear signals even at distances of several kilometers, and 5 minutes (37 total periods) with reduced power in case of overloads at nearby data loggers. The maximum injection current into the ground was 22.4 A with an average of 10.2 A for all injections. As voltage electrodes, non-polarizable electrodes (Ag-AgCl and Cu-CuSO$_4$) were used to avoid polarization effects over the current injection time. To register voltages, two data recorder types were used (24 RefTek Texan-125A single-channel recorder and 10 self-developed remote-controlled 3-channel data logger (Oppermann and Günther, 2018). A continuous registration of the full time series with a 100 Hz sampling rate for the single channel recorder and 200 Hz sampling rate for the 3-channel data logger was carried out during the survey to account for possible high-frequency noise signals. The field experiment is followed by comprehensive data pre-processing, including data storage, compilation of the raw data in a data base system, raw data quality analysis, and raw data processing.

## 3.2 Small-scale ERT survey

In preparation of the large-scale experiment, several near-surface surveys using a commercial GeoTom multi-electrode device were carried out in proximity to the large profile. Due to the specific setup of the large-scale experiment and the limited resolution within the first tens of meters, additional surveys with small electrode spacings provide useful information about the

near-surface resistivity. 100 steel electrodes with a spacing of 5 m were used in these surveys resulting in a total length of 495 m for a single profile. The setup is similar to the ERT profiles shown by Nickschick et al. (2015) and Nickschick et al. (2017) for comparison purposes. Thus, we also measured in Wenner alpha and Wenner beta configuration due to the good results from these previous studies. Both arrays have been combined and were inverted with the BERT software (see section 3.4) using a vertical-to-horizontal smoothness factor (Coscia et al., 2011) of 0.2, i.e., making vertical gradients five times more sensitive than horizontal ones.

## 3.3 Data processing of the large-scale ERT data

Natural and anthropogenic sources and industrial facilities near the the profile lead to noise within the acquired voltage time series. To reduce noise and eliminate unwanted signals, data processing is required. This issue was addressed by a signal enhancement procedure with a selective stacking approach from Friedel (2000). The approach aims at stacking the acquired voltage time-series $U(t)$ (Fig. 3a) into separate cycles.

The first step in the processing procedure is a drift correcting to remove the DC voltage parts and long-periodic drift components (Fig. 3b). This is realized by applying a filter function yielding the drift-corrected function $U_{dr}(t)$ that subtracts the moving mean value of the time series $U(t)$ with a window size of the injection signal period $M$ from the original time series $U(t)$, as suggested by Friedel (2000):

$$U_{dr}(t) = U(t) - \frac{1}{M} \sum_{d=-M/2}^{M/2} U(t+d) \tag{1}$$

This provides correct results in case of a symmetric signal with an identical positive and negative amplitude, which is given in this case by controlling the source and assuming that the signal is not distorted by having a very high signal-to-noise ratio. The next step is to reduce short-term noise. In this case, this is done by stacking the events using the $\alpha$-trimmed-mean-stack (Naess and Bruland, 1979; Friedel, 2000; Oppermann and Günther, 2018), in which every sample within the stacked signal period is sorted by amplitude and the smallest and largest amplitudes that exceed a portion of $\alpha$ are rejected. Here, we used a rejection rate of $\alpha = 10\%$, resulting into a mean that is less susceptible to outliers by removing the most deviating 10% of the samples. To determine the phase shifts between injection signal and registered signal, a cross-correlation between the stacked signal and an ideal waveform needs to be found. This is done by stacking at an arbitrary point and determining the phase of maximum cross-correlation. As a final step, the response time of the current switching (transients) before reaching the plateau has be considered. A window is selected that ignores an fixed amount of samples (typically 10 %) before and after the current switch. In the end, we get a stacked signal as seen in Fig. 3d. The voltage $U$ is the half difference between the positive ($U_p$) and negative ($U_n$) plateau voltages,

$$U = (U_p - U_n)/2. \tag{2}$$

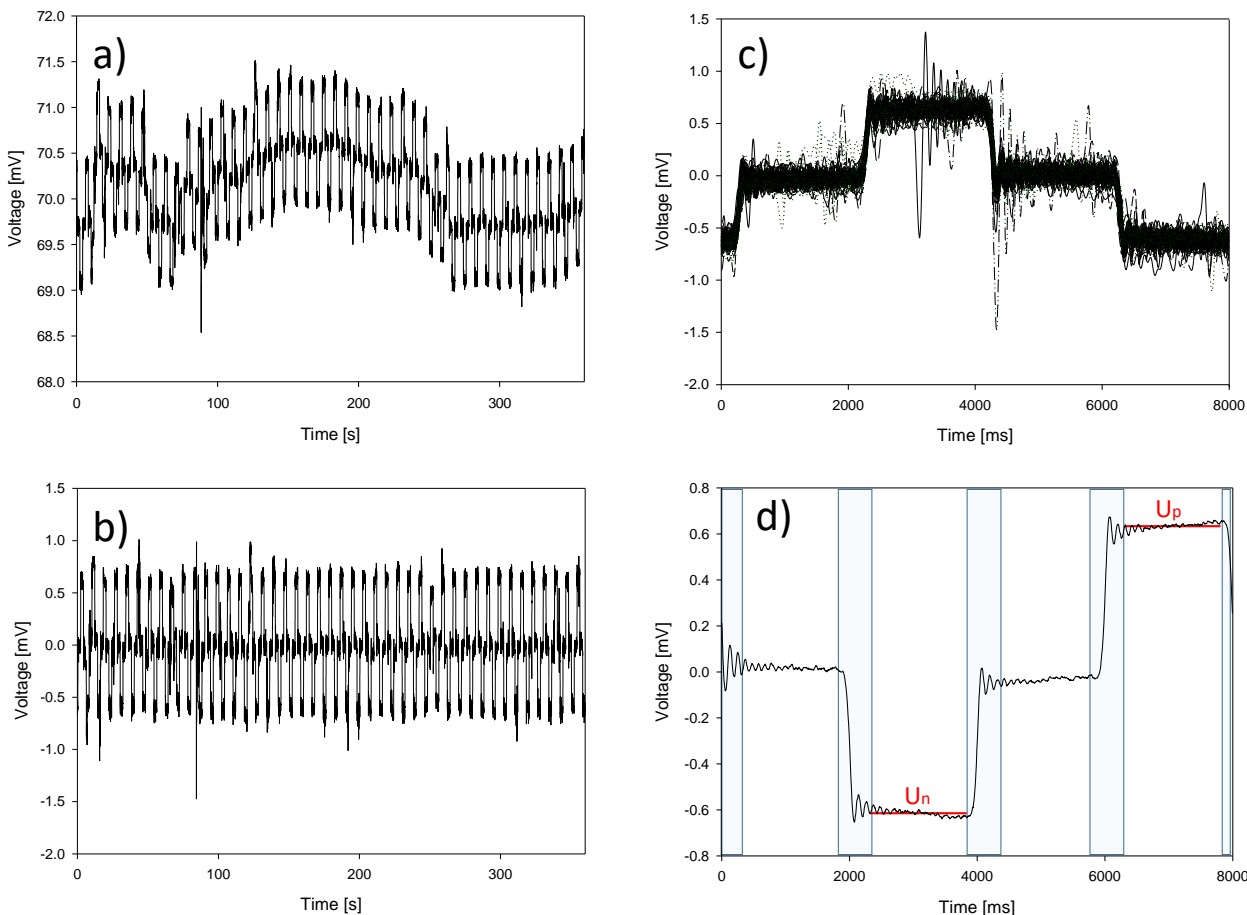

**Figure 3.** Processing steps of time series on an example. a) Raw time series $U(t)$, b) Time series $U_{dr}(t)$ after drift correction, c) Stack distribution after cross-correlation, d) Mean stacked signal with rejection windows to delete current switch effects (greyish areas) with positive ($U_p$) and negative ($U_n$) mean plateaus.

This has to be done for each of the 54 receiver dipoles at the 47 current injections, leading to a theoretical number of 2538 current-voltage pairs for this setup. However, this is reduced to a number of 2397 because voltage is not measured at the current electrodes.

In theory, every combination of current and voltage dipole is measured twice by taking into account the principle of reci-5 procity, which states that voltage and current can be interchanged. By comparing the apparent resistivity values for forward (AB dipole ahead of MN), $\rho_f^a$, with the backward (AB behind MN) values $\rho_b^a$ one can compute the relative reciprocity error

$$r = \frac{\rho_f^a - \rho_b^a}{\rho_f^a + \rho_b^a} \tag{3}$$

for each reciprocal pair. This value should be zero, but in practise it is not due to (i) different coupling of current injection fields compared to potential electrodes, (ii) individual noise levels at different voltage gains leading to different signal-to-noise 10 levels. Therefore it can be used as a measure of data consistence and also to derive error models (Udphuay et al., 2011), however only if a statistically large number of data is available.

Figure 4(left) shows the raw apparent resistivity $\rho_a$ cross-plot as a function of current and voltage dipoles, which should be theoretically symmetric. White areas are blank due to injections at the respective voltage reading positions (three inner diagonals), dominant noise in the time series or missing cable connection. In the few cases where the voltage was too high (e.g. 15 at neighboring dipoles), the smaller current injection was chosen to fill up the missing data. In all other cases, the injection with higher currents leads to better signal-to-noise ratios.

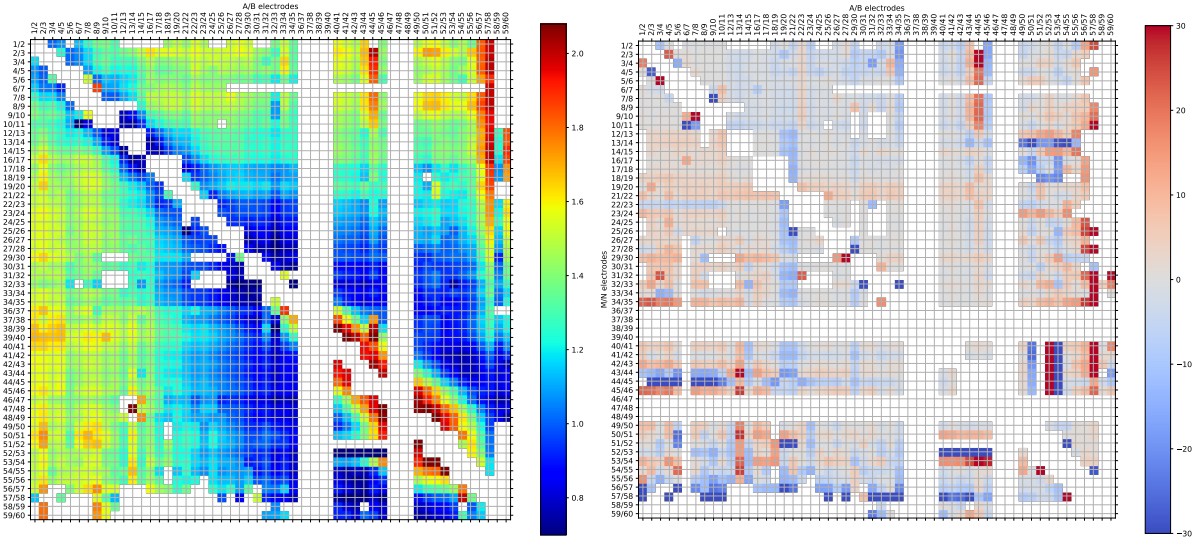

**Figure 4.** Raw data (all retrieved AB-MN pairs) as a function of current AB and potential MN dipoles. Left: Apparent resistivity (log $\rho_a$ [$\Omega$m]), Right: Relative reciprocal error between forward (starting with current dipole 1/2) and reverse measurements (%).

Many factors interfere with the experiment and the voltage readings, decreasing the amount of reliable data. Strong, irregular signals of 16.7 Hz superimpose the data record of the westernmost logger (1/2) which can be attributed to rail traffic 800 m

south of the western part (Fig. 1) of the profile and leading to a high artificial signal input in general. The easternmost voltage readings (58/59 and 59/60) are often overlain by anthropogenic signals from the village of Kaceřov. Furthermore, the current injections show a highly disturbed injection signal, which we attribute to a buried gas pipeline, as indicated by their appropriate sign in the vicinity. Therefore these data had to be removed. Some of the planned injection dipoles (35/36 to 38/39 and 47/48 to 48/49) could not be accessed with the trailer-mounted current source due to roadside ditches and high crop growth at that time. Fortunately, the missing data (white columns) are mainly available through their reciprocals.

The reciprocal error is displayed in Fig. 4 (right). A large portion of the area appears grey, i.e. forward and backward data agree very well. For some data with short spacing (near the diagonal) the values deviate from zero due to different coupling. In general, reciprocal errors increase with increasing dipole separation and reflect the decreasing signal-to-noise ratio as a results of the strongly decaying signal strength.

In order to appraise the quality of the partially redundant sub-datasets, we used both data in preliminary inversion runs using default parameters. It turns out that the RMS misfit of the upper-right triangle was explicitly lower (11%) compared to the lower-left triangle (27%) which shows more systematic structures in the misfit plot. Therefore, we decided to fill up missing data in the further by the latter. The further workflow has the aim of generating a homogenized pseudosection. It consists of the following steps (cf. Oppermann and Günther, 2018)

– removing bad data (single outliers visible as point or point groups such as the aforementioned AB pair 44/45),

– filling the missing values in the upper right triangle with the corresponding data in the lower left triangle,

– computing the data reciprocity for the doubled data from the resistivity,

– replacing the corresponding resistances by the current-weighted mean of the two.

As a result, we obtain an apparent resistivity pseudosection as known from multi-electrode measurements (Fig. 5 left), i.e. plotting the value as a function of the midpoint position and the separation (dipole distance normalized by dipole length).

For small separations, we observe low values (5-20 $\Omega$m) in the west, and higher values (40-200 $\Omega$m) in the east. The apparent resistivity increases with separation, which is more pronounced in the western part. There are still two white stripes for a dipole with a missing registration.

## 3.4 Modeling and inversion of the resistivity data

The aim of the inverse modeling is to find a subsurface resistivity distribution that is able to reproduce the measured data. We use a smoothness-constrained Gauss-Newton inversion (Günther et al., 2006) implemented in the freely available software BERT (Günther and Rücker, 2009). The whole data processing and visualization uses the pyGIMLi framework (Rücker et al., 2017) in Python. The subsurface is discretized by triangles so that the measured topography can be taken into account accurately. The maximum model depth is determined by 1D sensitivity analysis with about 130 m for the small profiles and 1300 m for the long one.

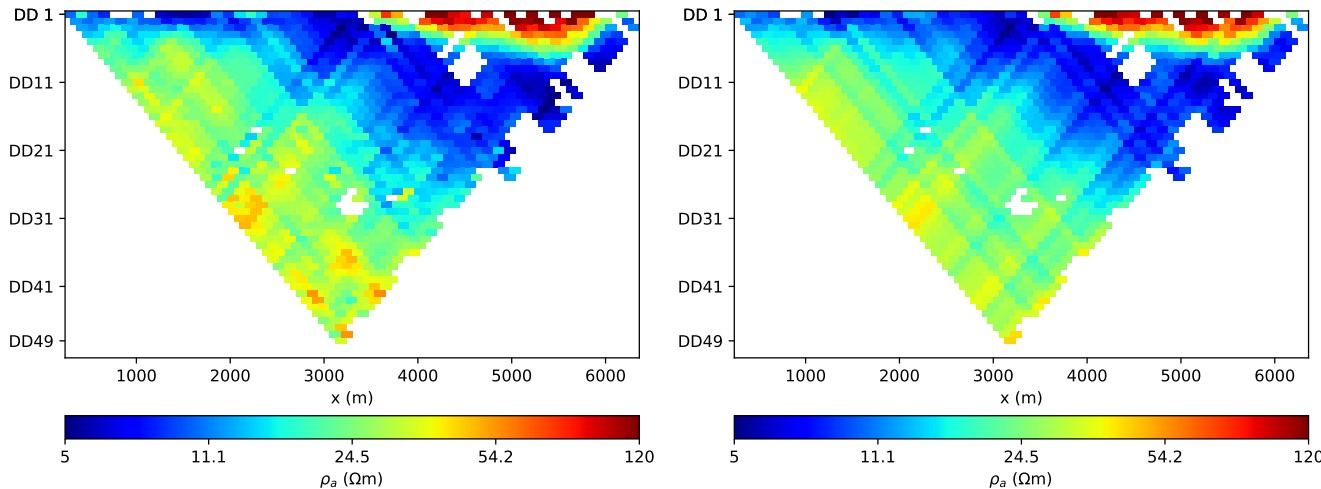

**Figure 5.** Unified data set as apparent resistivity pseudo-section: measured data (left) and model forward response (right).

In the inversion process, the individual data points are weighted by error estimates consisting of a percentage error and a an absolute voltage error so that measurements with lower voltage gain have less importance than those with strong signals. Reciprocal data can be analyzed statistically in order to obtain numbers for this error model (Udphuay et al., 2011). In our case, we determined a percentage error of 5% and a voltage error of 2 $\mu$V, leading to maximum error estimates of 20% maximum

for the large-scale ERT data's weakest signal at maximum distance. For the small scale ERT profiles, no reciprocal data were available so that we used the default values of 3% plus 100 $\mu$V.

For the regularization, we used smoothness constraints of first order as described by Günther et al. (2006). However, to account for predominantly layered structures (larger correlation length in x direction compared to z direction), we applied a vertical smoothness factor (see Coscia et al., 2011) of 0.1, i.e. purely vertical gradients in the model are ten times less penalized

than purely horizontal gradients. The overall regularization parameter (300) was chosen such that the data were fitted within the estimated noise level, i.e. with a chi-square error (root mean square of error-weighted misfit) of about 1. Whereas this corresponds to RMS values of about 5% for the short profiles, the large profile shows a relative misfit of about 12%.

The forward response, i.e. the apparent resistivity theoretically measured over the retrieved resistivity subsurface, is displayed in Fig. 5. One can see that the main structures are reproduced by the model, but not the detailed outliers due to error

weighting, resulting in the overall misfit of 12%

### 3.5   Gravity survey

In conjunction with the resistivity survey, we also measured gravity along the ERT profile in order to have additional geophysical data for interpretation. For this purpose, a LaCoste & Romberg D-188 gravimeter was used for gravity surveys in 2017 along the ERT profile. Its resolution is 0.001 mGal and we achieved an accuracy of 0.006 mGal. In the central part of the

profile, very detailed measurements from the previous investigation of the Hartoušov degassing zone from 2012 on profile 2 from Nickschick et al. (2015) were included. To double-check the accuracy of the new surveys in comparison to the older one, several points from that profile were located and re-measured. The average difference was only 0.008 mGal. The spacing on the profile between each measurement station was 40-60 m, while the spacing in the central zone on this profile is denser (10–40 m). Thus, a total of 170 stations exists along the profile. The gravity measurements were referenced to the Czech national gravity network. All essential corrections were applied (drift, tidal, latitude, free-air, Bouguer, terrain). Coordinates were observed by Trimble R9 RTK technology, the accuracy of all these measurements was better than 0.03 m in vertical component. Terrain corrections were calculated from an accurate digital elevation model (DEM) of 1 m resolution to the distance of 250 m, the outer part of the correction to 167 km from the SRTM90 DEM. As the profile was located in the Cheb Basin, the reduction density of 2300 $kg * m^{-3}$ was applied for computing the Bouguer anomalies.

## 4   Results

### 4.1   Small-scale ERT

The three short ERT profiles (625-700 m long) provide insight into the uppermost (approximately 100 m) resistivity distribution along the large-scale profile (Fig. 6). We chose an identical colorscale for all results that helps comparison with the large-scale profile. We used the coverage (cumulative sensitivity of the final model) for the alpha-shading of the inversion results so that poorly covered model regions will not be interpreted. We refer to Ronczka et al. (2017), who substantiate the use of coverage as a simple approximation for the model resolution by using synthetic and field ERT data.

Profile 1 (Fig. 6 top, final RMS after 3 interations=4.7 %), located in the western part of the large-scale profile, reveals that the first 5 meters of this profile feature resistivities of less than 100 Ωm. This layer is on top of a rather massive and homogeneous compound of conductive rocks which is characterized by resistivities of 15-60 Ωm between 5 and 20 m depth, and an even more conductive (<15 Ωm) zone beneath.This resistivity distribution encountered here fits into the geological description of drilling B-18. The first few meters consist of resistive Quaternary sand and loam compared to the lower resistivity that is the underlying Cypris formation. The drill log describes the area beneath 20 m as water-saturated so it can be assumed that the first 20 meters are not saturated and thus slightly less conductive.

Profile 2 (Fig. 6 middle, final RMS after 9 iterations=17,4%), crossing the mofette field Hartoušov, confirms the findings from Nickschick et al. (2015): a resistive (>150 Ωm) layer of ca. 15 m thickness can be measured on top of the more conductive zone. At approximately 400 m profile distance, just as the elevation increases towards the east, a significant thickening of the high-resistivity near surface layer can be observed. The resistivity distribution in the western part of the profile 2 fits the description of drilling SA-30 and the new drilling HJB-1 (Bussert et al., 2017): the first 15 meters consist of gravel, sand and peat, resulting in overall higher resistivities compared to the Tertiary sediments below. Discrepancies in the core description between drills SA-30 and HJB-1 reveal that deposits (clay and gravel) from the Vildštein formation are found in the area. We link the sudden shift in resistivity and elevation from 400 m onward to be linked with the increased thickness of the Vildštein deposits towards the East, as stated by the drill logs. This sudden and sharp lithology shift is linked to the course of the PPZ

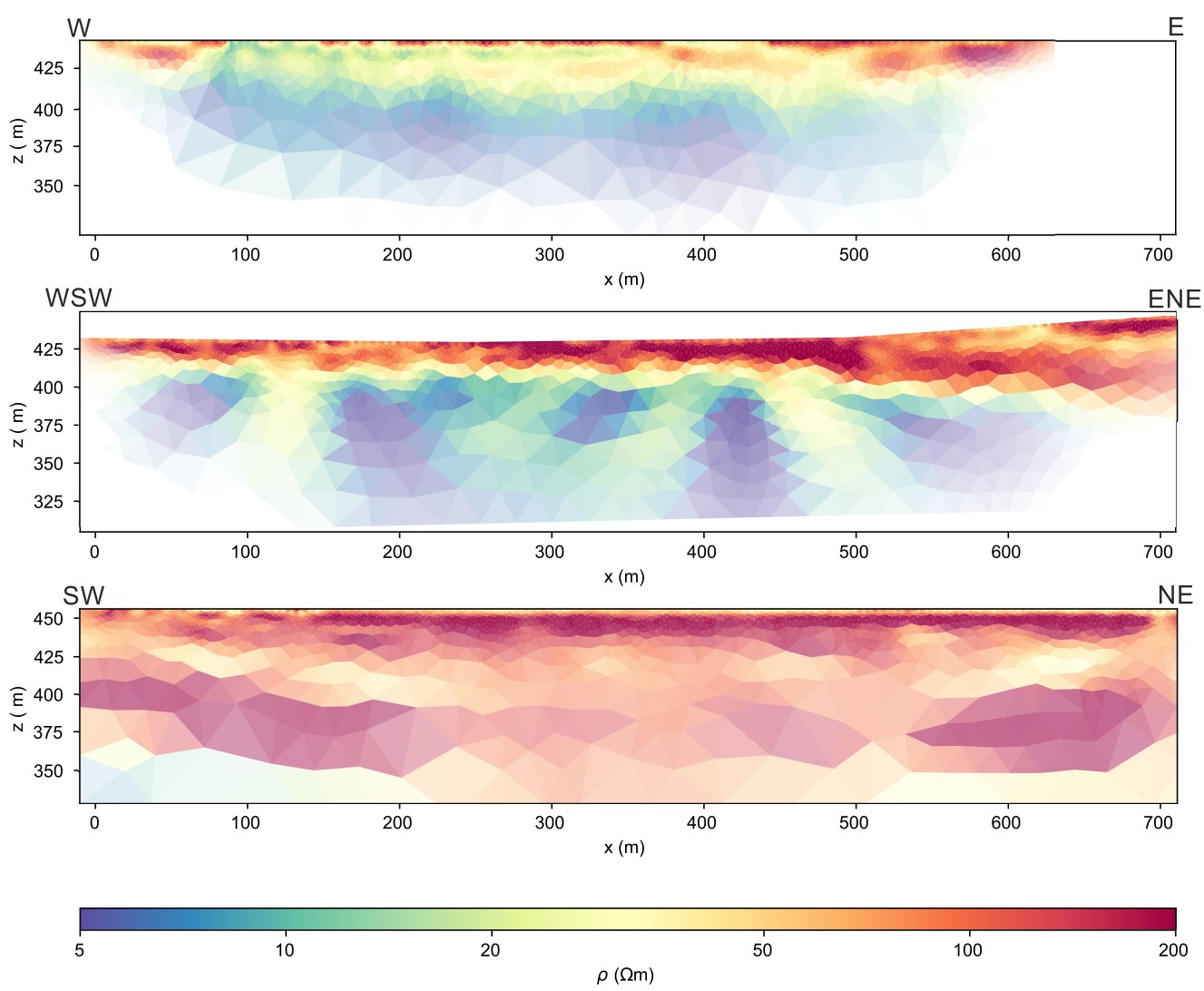

**Figure 6.** Resistivity distribution of the small-scale ERT profiles: 1 (top), 2 (middle), and 3 (bottom), z: m.a.s.l., (s. Fig. 1 and 2 for locations and lithology). Transparency represents the coverage (cumulative sensitivity) and helps avoiding interpretation of uncertain model parts. The relative RMS values are 4.7%, 17.4% and 4.5%, respectively.

and vertical offsets of a few tens of meters due to various stages of subsidence and lifting (Bankwitz et al., 2003a; Peterek et al., 2011; Kämpf et al., 2013; Rojik et al., 2014; Nickschick et al., 2015). It is to be noted, that the vertical plume-like anomalies could be linked to areas of strong $CO_2$ degassing at surface as reported in previous studies (Flechsig et al., 2008; Nickschick et al., 2015, 2017).

Profile 3 (Fig. 6 bottom, final RMS after 14 iterations =4.5 %) reveals a 10-15 m thick layer with resistivities above 300 Ωm on top of a massive compound of rocks with about 150 Ωm, which is significatly higher than in profiles 1 and 2. At about 100 m depth, resistivity decreases, but this represents the investigation's depth limit.Core descriptions from nearby drills, such as B-1 or SA-31, indicate a 10-12 m thick layer of Quaterny deposits as the topmost layer. Clayey and silty-sandy Vildštein deposits, however, have reached thicknesses of 60-80 m in this area according to the core descriptions, which reflects in higher
resistivities compared to the very conductive Cypris formation at the bottom.

## 4.2   Large-scale ERT profile

Figure 7 shows the inversion result of the long profile after 4 iterations (RMS evolution - 81%, 29.9%, 18.5%, 14.6% and 14.5% as the final RMS). On top, the lithology provided by the neighboring drills is plotted as colored boxes columns, indicating the limited depth that has been achieved by the drills.

The 2D-resistivity distribution of the profile shows remarkable differences in the structural composition in the western half of the profile compared to the east half. We observe a well-conducting layer of <30 Ωm of about 200 m thickness above a basement of higher resistivity (>100 Ωm) in general. The transition is gradual. At about 2500-3000 m along the profile, these layers dip towards the east and form a trough-like structure before ascending again upwards to the eastern end. This also leads to the occurrence of another layer of >100 Ωm at the surface between 3200 and 5800 m which reaches a maximum thickness
of about 300 m. The lowest resistivities (5 Ωm) are found along 4000-5000 m along the profile at a depth of 300-500 m.

## 4.3   Gravity

The gravity survey (Fig. 7 bottom) reveals a total maximum relative gravity difference of about 9 mGal along the profile between the local maximum at ≈1500 m and the minimum at 6300 m. It is to be noted that the gravity minimum is measured at the point of highest elevation. The maximum is located where a high-resistivity anomaly is observed in the profile and
the minimum is slightly west of the area where the lowest resistivities were measured. The slight shift between these two observations might be related to the different sensitivity of the electric resistivity and density towards changes in the lithology in north or south of the profile. This gravity trend is enhanced by the W–E trending contact of phyllitic (on the southern side) and granitic (on the northern side) rocks in the basement, according to Hecht et al. (1997). The central section around the Hartoušov moffette field is located on the crossing of this zone with the Počátky-Plesná fault zone and the gravity gradient
delineating the deepest part of the basin. Such tectonic/structural zones form permeable channels for the deep fluids conduct and have been mentioned before for this area Bankwitz et al. (2003b); Kämpf et al. (2013); Bräuer et al. (2008); Fischer et al. (2014, 2017). In Nickschick et al. (2015), we proved that detailed microgravity measurements in the mofette area is capable of locating particular small-scale degassing channels due to decreased bulk density of the rocks, which are in the range of a few

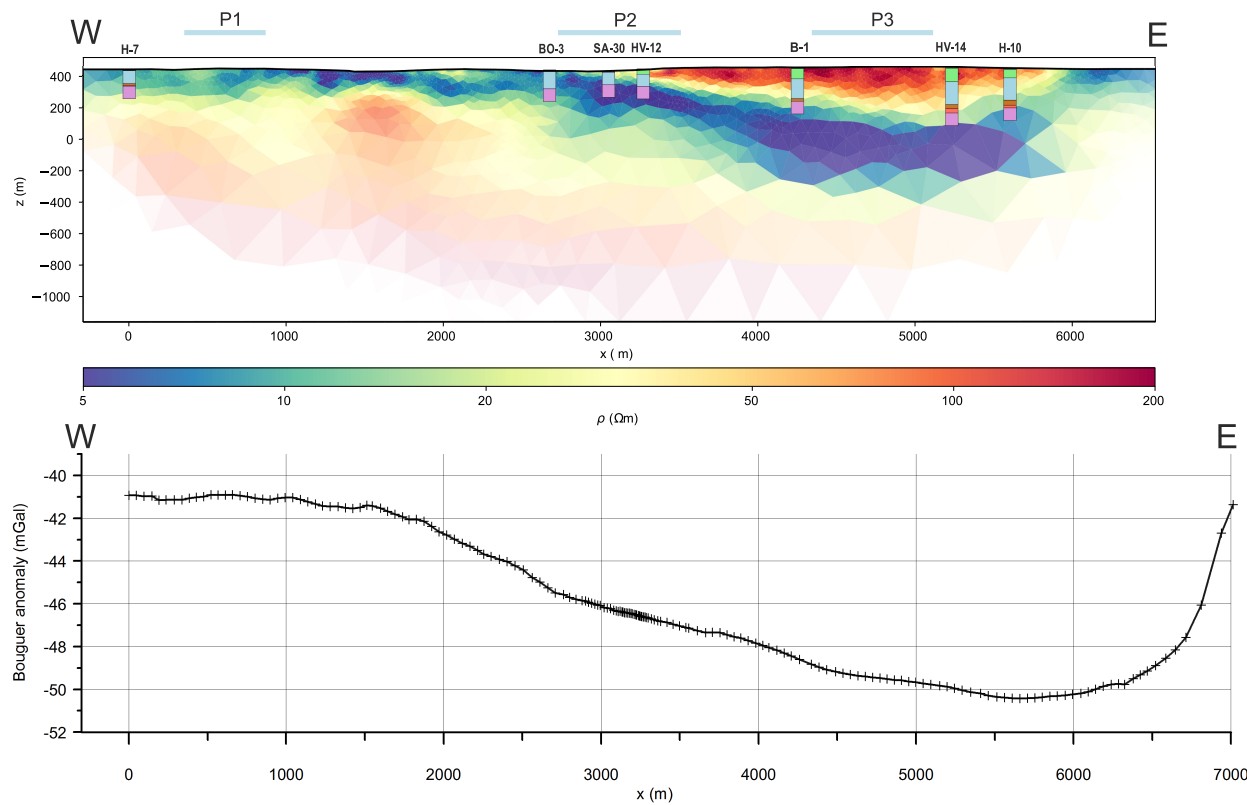

**Figure 7.** Inversion result (resistivity distribution) of the large-scale profile with the lithology columns of the boreholes (top) and the Bouguer gravity (bottom). z: m.a.s.l.. Colors for each stratigraphic unit is identical to Fig. 2: green - Vildštein formation, light-blue - Cypris formation, brown - coal, red - Lower Sand formation, pink - phyllitic/granitic basement. The relative RMS is 14.5%, coverage-based alpha-shading is used as in Fig. 6.

tens of microgals and thus not visible on this scale. At the eastern end of the profile, gravity increase indicates the contact of sediments with outcropping basement of the Krušné hory Mountains.

## 5 Interpretation

Using available drill logs from the Czech Geological survey, we can interpret the upper part of the resistivity distribution as lithologic units: The topmost few meters are generally marked by a high resistivity layer and relate to Quaternary deposits, mainly gravel and sand, as described in these logs. This layer is, due to its low thickness, only visible in the near-surface ERT results (Fig. 6). We can clearly relate the high-resistivity zone between 3200 and 5800 m to the deposits of the Vildštein formation with the help of the drill core descriptions. The higher amount of silt and sand results in a higher resistivity compared to the underlying Cypris formation, whose higher portion of clay minerals results in the overall well-conducting layer and provides a rather sharp contrast. The transition to the basement is, however, not well-defined: Most of the existing drill core

and borehole data only provide information up until the base of the Cypris formation or, in the eastern part, until the coal/lignite and Lower Sand Formation has been reached (Fig. 2). Stratigraphic records mention the occurrence of phyllite at the base, yet it is described to be very heavily weathered/altered (Dobeš et al., 1986; Špičáková et al., 2000; Fiala and Vejnar, 2004; Bussert et al., 2017).

As mentioned before, reliable data on the thickness of the weathering zone itself and the transition to unweathered phyllite are scarce. To our knowledge, only one drill in the vicinity provides sufficient information for depths >0.5 km: borehole HV-18 (E:314979, N:5553582 in UTM 33N) with a total depth of 1200 m and well-described by Fiala and Vejnar (2004) and Dobeš et al. (1986). From this drill hole we can infer that underneath the compound of Tertiary deposits, different types of phyllite/mica schist occur. It is described as mostly normal phyllite with varying additional horizons of tuffitic, silicified,
metabasite-bearing or $FeS_2$-bearing layers (Dobeš et al., 1986). The petrophysical measurements on core and outcrop samples reveal resistivities of over 500-1500 $\Omega$m for slightly weathered to unweathered phyllite (Tab. 1) which we do not observe in our survey even in the deepest parts. Dobeš et al. (1986) also mention the high variability of the thickness of the weathered phyllite within the Cheb Basin but is assumed to be within several tens of meters which is characterized by resistivities of 75-140 $\Omega$m. It is to be noted that these values are higher by one to two orders of magnitude than the resistivities in the Tertiary sediments.
While the sediments of the Cypris formation are characterized by porosities of 21.2% for the porous sandstone parts and 14.5% for compact carbonate layers, the basement phyllites are characterized by low porosities ($\approx 3.2\%$ for weathered phyllite and 1.0% for unweathered phyllite, Tab.1). However, our experiment reveals low-resistivity rocks of only 5-10 $\Omega$m up to several hundred meters of depth - much lower than expected from these previous studies. A similar phenomenon was also presented by Muñoz et al. (2018) in which a N-S running magnetotelluric survey reveal an unusually conductive zone within the topmost
kilometer beneath the degassing centers of Bublák and Hartoušov. This observation also makes the interpretation of the gravity data significantly harder. While, generally speaking, the Tertiary deposits should feature a distinct density and porosity contrast compared to a solid basement, the assumption of a massive compound of weathered/alterated phyllite - and the induced density shift - in between makes a gravity-based model without further constraints near impossible.

    One key aspect in the low resistivities we observe (see Fig. 6, profile P2), might be related to circulation and ascent of
25 heavily mineralized water and $CO_2$-rich fluids. Bussert et al. (2017) mention pumping tests at the HJB-1 drill site within the main degassing area around Hartoušov and, after drilling through a caprock-like layer and hitting a supposed aquifer at 79-85 m, encountering subthermal mineral water with a high conductivity of around 6800 $\mu$S cm$^{-1}$ (about 1.5 $\Omega$m). Especially the more porous sandy parts within the Tertiary deposits are aquiferous and penetrating them resulted in a sudden outburst of gaseous $CO_2$ and water (Bussert et al., 2017). While especially the pelitic layers can be considered impenetrable to groundwater,
intense tectonic faulting is made responsible for the mixture of groundwater with deeper water-bearing formations along faults, joints and chasms and also with the aquiferous Lower Argillaceous-Sandy and Main Seam formations (Dobeš et al., 1986; Peterek et al., 2011; Bussert et al., 2017). This is stressed by geoelectric borehole logging in the HJB-1 drill at the HMF where throughout the Tertiary sediments resistivities of 5-10 $\Omega$m were measured and even within the topmost layers of the (weathered) basement (phyllite) resistivities did not exceed 20 $\Omega$m. Another, prominent example for the complexity of the
hydrologic situation is the close-by Soos Nature Reserve, which is just about 3 km to the NW of our survey profile (Fig. 1.

Other mineral and ochre springs and mofettes are found within a few kilometers (Weinlich et al., 1998; Bräuer et al., 2005; Kämpf et al., 2013) and Karlovy Vary, Františkovy Lázně, Mariánské Lázně, Bad Brambach and Bad Elster are well-known for their spas and diverse mineral water sources.

Our survey shows that even within the Cypris formation resistivities vary, depending on the hydrogeological situation. Especially in the eastern half of the profile where the basin deepens, we observe higher resistivities than in the western half. One major key factor could be the absence of circulating mineral water in the sedimentary deposits in this part of the region due to a lack of tectonic faults. Instead, the lowest resistivities can be measured underneath in the phyllitic basement, indirectly implying an unusually high porosity or fractures within the basement and the occurrence of ion-enriched water in pelites, which are supposed to be be compact and rather dense. Several studies (Fiala and Vejnar, 2004; Špičáková et al., 2000; Rojik et al., 2014; Peterek et al., 2011; Bankwitz et al., 2003b) provide indications for heavy strain of the Paleozoic basement. Especially the intrusion of the Smrčiny pluton in the Carboniferous, whose contact zone to the phyllitic basement is close to our profile, and the rifting of the Eger Rift since the early Oligocene (Ziegler, 1992; Ziegler and Dezes, 2007) with several extensional and compressional stress regimes have lead to alterations and faults in the basement. These studies all show a basement that is heavily distorted by horsts and grabens and it can be assumed that at least some of these provide preferential pathways for mineralized and $CO_2$-rich water within the upper crust. Along our profile at the HMF, these fluids can propagate to the surface through the Tertiary sediments along the PPZ, but also at other sites expressions of fluid flow can be observed (Weinlich et al., 1998; Kämpf et al., 2013; Bräuer et al., 2014). In addition, the E-W running contact zone of the Smrčiny pluton with the crystalline basement itself has been assessed as a major migration path of juvenile $CO_2$ (Dobeš et al., 1986, and articles therein). One striking feature in our survey is both the gravity and resistivity anomaly between 1500 and 2000 m along the profile at a depth of >200 m. Since other authors (e.g. Dobeš et al., 1986; Fiala and Vejnar, 2004; Špičáková et al., 2000; Pešek et al., 2014) also mention local basaltic effusiva at the base of the Tertiary deposits, a possible explanation might just be the existence of such an intrusion at this point. Another hypothesis could be a rather substantially lifted block of the basement due to tectonic compression. Most tectonic-based publications (Špičáková et al., 2000; Bankwitz et al., 2003b; Peterek et al., 2011) discuss the occurrence of multiple N-S running faults in the Cheb Basin, such as the PPZ or the Skalná fault. Bankwitz et al. (2003b) and Peterek et al. (2011) mention the so-called Lužni fault as N-S striking, 1 km to the east of and parallel to the PPZ, whose presence is derived from drainage patterns and the course of the Lužni brook and Sázek river. The projection of this fault onto our profile coincides with the resistive anomaly we measured. However, a potential fault in this case would rather lead to a negative gravity anomaly and not the positive one that is observed.

Clear limitations of this experiment occur at depths where stratigraphic records do not exist and thus do not provide essential information, but several assumptions can still be made about the relevant parameters.

1. Clay content: Basically the whole basin and basement is dominated by clay minerals: The Vildštejn formation has a very high, but varying clay content, the Cypris basin is made of mudstone and the basement consist of phyllite/claystone/mudstone, which are weathered and re-mineralized as clay minerals.

2. Saturation: the groundwater level overall is very high, ranging from about 5m in the center to a few tens of meters at the eastern and western flank, otherwise the clay-dominated rocks can be considered as saturated and considering our resolution of 100 m, this transition from unsaturated to saturated is barely noticeable in the large-scale profile. The separation of free $CO_2$ from its solved state in water is interpreted at about 50 m, according to findings from Nickschick et al. (2015), but heavily depends on pressure.

3. Porosity: It is nigh impossible to make specific statements about the porosity at a depth >300 m due to the complex interaction of lithostatic pressure, hydrostatic pressure, pore size change by mineral alteration, mobilisation and precipitation from the aggressive fluids and general occurrence of joints and chasms by the earthquake activity and the p-T conditions in this geodynamic setting. For none of those, information can be found for the lower two thirds of the model.

4. Fluid conductivity: This is can also be assumed as highly variable. We have one single measured value for the mineral water conductivity at the central mofette site (6800 $\mu$S cm$^{-1}$ Bussert et al. (2017) for this specific area. However, we have to consider groundwater, various different mineral water springs, $CO_2$-bearing water (dissolved $CO_2$ as well as a high fraction of solved ions) free gas near the surface, supercritical $CO_2$ at some hundreds of meters of depth which change depending on the location and depth and their aquifers.

Studying these in the future will be interesting when additional data might be available from the ICDP drilling or seismic exploration. Until then, the high variability of each parameter can only be estimated very roughly with this large-scale experiment and its resolution.

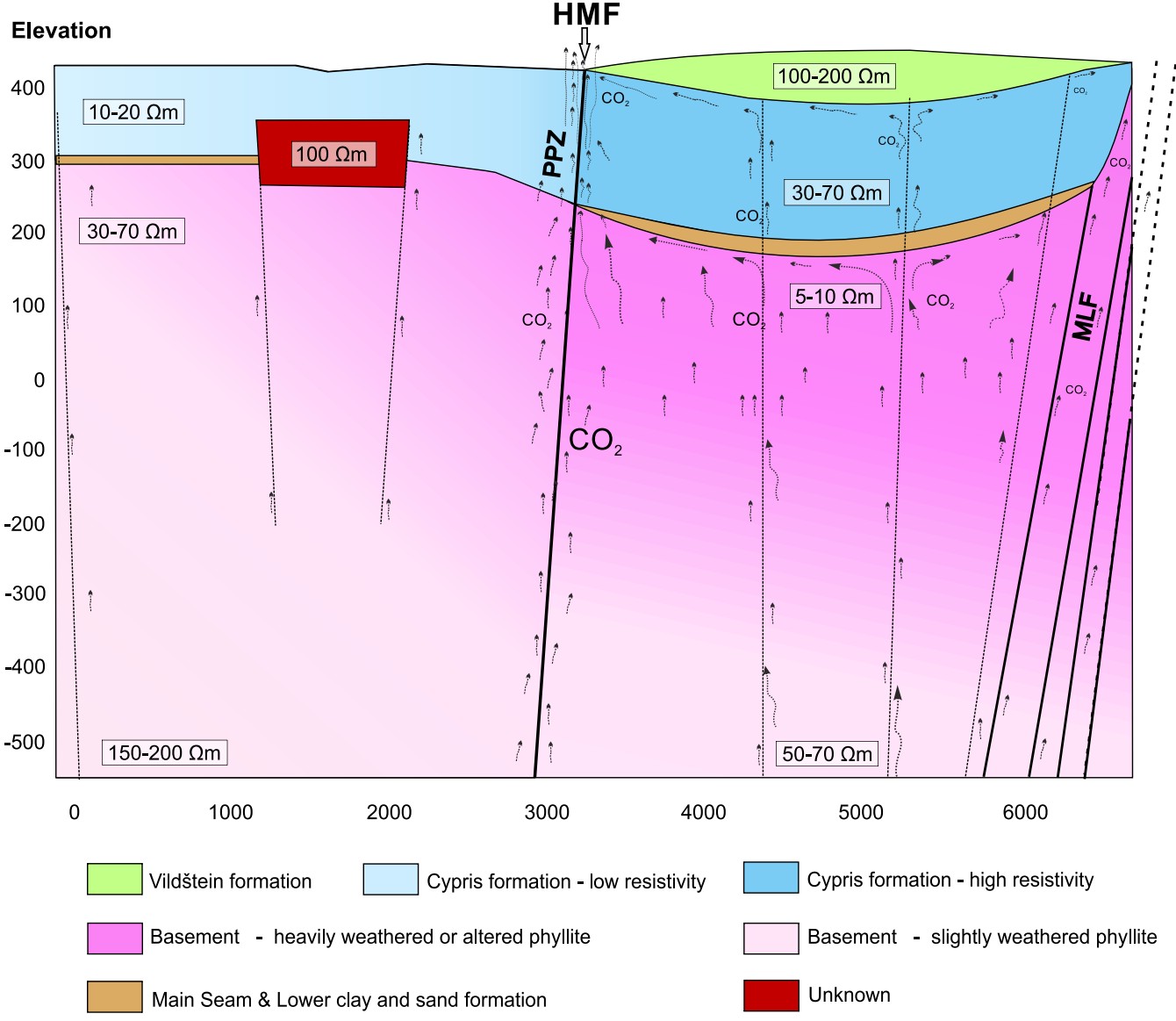

**Figure 8.** Conceptual W-E model of the topmost 1000 m of survey area. Stratigraphic units are based on drill core information from Fig. 2 for the first 100-300 m. While the Cypris and Vildštein formations are characterized by higher resistivities in the eastern part, the basement features lower resistivities, which is attributed to $CO_2$ ascent, rock alteration and highly mineralized water. Dashed lines are inferred faults. The first few meters of Quaternary coverage are not shown.

In Fig. 8 we combined our findings from this survey and existing lithologic information for the topmost 600 m. We can observe less-resistive Tertiary deposits in the west than in the east, which we link with the occurrence of the rather thick Main Seam and Lower Sand and Clay formations working as a cap for the ascending fluids in the eastern part. On the other hand, the

basement features very low resistivity in the (weathered) basement in the eastern part. Assumed pathways in the conceptual model are derived in the conceptual model from the facts, that

- Ascending, magmatic CO2 and fluids are migrating to the surface through the basement and basin sediments,

- Existing faults are used as preferential pathways (PPZ and MLF as confirmed faults, several subfaults as mentioned by Špičáková et al. (2000) and Pešek et al. (2014) with mofette activity on top, and

- The eastern side of the Tertiary sediments appears to be dry compared to the western side, while it is the other way around within the basement, indicating that the lignite/lower Clay and Sand formations act as a fluid trap for the ascending fluids which forces accumulation and hinders further ascent.

Due to setup and resolution limits, including additional data in form of stratigraphic records would provided valuable informa-
10 tion especially for these cap-like formations by clearly distinguishing stratigraphic units from changes in the electric resistivity and determining the very thick 500-600 m unit of low resistivity rocks which are interpreted as heavily weathered or alterated, phyllitic basement.

## 6 Conclusions

Our field survey aimed at imaging the fluid-related or fluid-affected conductivity structures beneath the Hartoušov mofette field
(and its surroundings), the most prominent degassing site and center of future and present drills in the Cheb Basin. Especially the planned 400 m ICDP drilling (Dahm et al., 2013) and the related fluid and microbiology studies will have to account for these results. Previously, it was thought that only the Tertiary sediments are significantly influenced by the water/$CO_2$ mixture. Instead, we showed that even the basement seems to be very reactive towards the chemical and physical alteration caused by these fluids - not only the first tens of meters, but rather a few hundreds of meters. This also means that the electric resistivity
can vary significantly even within one stratigraphic unit. However, we were not able to find a distinct fluid channel at depth in the large-scale experiment. This might be related to the setup and resolution issues as we can trace fluid-related resistivity changes in the small-scale ERT profiles at the HMF. We are also not capable of finding direct evidence for the existence of the PPZ, but based on previous statements from Bankwitz et al. (2003b) and Peterek et al. (2011) and their estimations of only up to 30 m of vertical shift, we cannot hope to see it from resistivity observations alone at depth. It is possible that currently
undetected, diffuse gas emissions might occur also further to the east and west. Further, additional deep-reaching investigations (e.g. seismics) are needed to substantiate our interpretations and to obtain more insight into the $CO_2$ pathways, potential rock alteration and the subsequent influence on the petrophysical parameters (if so resistivity and density).

*Data availability.* Data available through ZENODO (ADD LINK). Contained are the readily processed data, not the time series, in the unified data format plus the BERT configuration

*Author contributions.* T. Nickschick and Ch. Flechsig planned the survey. T. Nickschick processed a large part of the time series, did inversions and interpretation. C. Flechsig is the PI of the project and helped with background and interpretation. F. Löbig did the small-scale ERT in his M. Sc. project and constructed the geological section from boreholes. F. Oppermann processed a part of the time series. T. Günther did the analysis of the processed data including inversion. J.Mrlina acquired and processed gravity data along the profile, and prepared gravity map. All authors helped in the field and wrote essential parts of the text.

*Competing interests.* The authors declare that they have no conflict of interest.

*Acknowledgements.* We would ike to thank the other members of the field crew (Robert Meyer, Dieter Epping, Vitali Kipke and Michael Grinat from LIAG Hannover, Theresa Rein, Helen Melaku, Andreas Lenz, Lutz Sonnabend, Roland Hohberg and Rene Voigt from Leipzig University, Vaclav Polak (IG CAS Praha) for GPS measurements during gravity survey, as well as Claudia Schütze from UFZ Leipzig) for their enthusiastic work under challenging field conditions. We also would like to express our gratitude to Marceau Gresse and two more anonymous referees who helped to impreove this article with their helpful remarks and suggestions. The joint project was funded by the German Research Council (Deutsche Forschungsgemeinschaft - DFG) under the grants FL271/16-1 and GU1095/5-1.

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
