# Peer review of "Large-scale electrical resistivity tomography in the Cheb Basin (Eger Rift) at an ICDP monitoring drill site to image fluid-related structures"

_Solid Earth, 2019_

## Referee Comment (RC1) · Anonymous Referee #1 · 19 Mar 2019

This is a review of "Large-scale electrical resistivity tomography in the Cheb Basin (Eger Rift) at an ICDP monitoring drill site to image fluid-related structures", by Nickschick et al. that aims to use geophysics to image fluid relevant structures in deep formations. This is an interesting application of a rarely-applied deep electrical imaging method and seems to be within the scope of the journal. The manuscript is written in acceptable English and the figures are well drafted, though several of the figures could be combined. The organization is adequate, but could be improved. Given that the claimed focus of the work is to elucidate fluid-related structures, I find that

there is relatively little treatment of this subject in the interpretation and discussion. Specific comments related to each of these general observations may be found below. I recommend that the manuscript be returned to the authors for revisions.

General comments: 1) strengthen the interpretation and discussion of fluids, or recast the purpose of the work towards structures (or whatever else seems most appropriate). 2) combine figures as noted 3) reorganize the text as noted, specifically focus on making the introduction flow better, ensuring that all content is in the appropriate section, and shortening the background section 4) given my comments below related to the complexities of interpreting ERT data due to convolved signals from porosity, chemistry, saturation, and clay, I suggest adding a focused section to the discussion (or interpretation) section clearly explaining how you tease apart these elements in your data.

Specific comments:

Introduction: the structure of the introduction is awkward, particularly because it immediately jumps into site description, without giving any big-picture setup or explanation.

Page 1, L12: "series of open questions" Either state those questions here, or move this text to where the questions are stated.

Page 1, Line 20: Change "drills" to "drilling"

Page 2, Line 33-34 & Page 3, Line 1: While I certainly agree that ERT is sensitive to fluids as indicated here, this justification for the ERT method seems incomplete because the several earth properties that control resistivity can be difficult to tease apart to attribute. As indicated on P2L34, the measurement is sensitive to porosity, salinity, saturation, and clay content all at the same time, and therefore the only way to retrieve any one of these parameters is to know three others.

Section 2.1 is very long and covers a wide variety of topics. Readability would be improved if this section was shortened and focused specifically on the topics most

related to the manuscript.

P6L32 – P7L1: Suggest breaking this into two sentences.

P7L3: Suggest to add a reference to support this statement on low resistivity areas.

P7L4&5: The topic of MT surveys was introduced back on Page 6: This text here seems repetitive, I suggest to reorganize or reword.

P8 L2: suggest to delete: "imaging a pathway from"

P8 L6-8: This text seems out of place. The authors have used this section to explain existing data, however this short paragraph indicates availability of data and explains their method for using it but does not explain the data. Could be rewritten to be more appropriate for this section, or moved to methods.

P8-L10-12: As indicated above, the nature of ERT interpretations is that these several properties all affect the measurement together, and therefore it is difficult to point to any one contributor as the primary control on electrical properties. Large porosity could have the same effect as high conductivity fluid in small pores. Low saturation could have a similar affect to small porosity. I think it is inaccurate of the authors to say "ERT is qualified for the detection of fluid signatures" without carefully explaining this statement in the context of how each material fraction contributes to the measured electrical signals.

P8 L20-21: ". . .for practical and theoretical reasons, most suitable for large-scale ERT experiments. . ." Please explain why, related to both practical and theoretical reasons. This seems like an important element of this manuscript given that such large scale measurements are so uncommon. It is also counterintuitive since Dipole Dipole configurations are well known to have poor signal to noise in comparison with nested arrays, for example.

Figure 2: I suggest either merging this with Figure 1 or Figure 3 to make a 2-panel figure, OR perhaps merging all three to make a single 3-panel figure.

Page 9, L2: "greater distances" suggest to replace this with actual distance numbers.

Figure 4 seems unnecessary and could be deleted.

Page 11, Line 9: Please deleted "A number of"

Page 12, L22: "Figure 6" Which panel of Figure 6 is being described here?

Page 14, L2: "(White Columns)" what does this refer to? Which figure?

Page 14, L3-4: What figure does this refer to? I assume #6. "appears significantly smoother" Smoother than what? How do you know it is "significant"? If referring to Fig 6, left panel, then I disagree – if the authors intend to make this argument, then it should be supported by a quantified metric.

Figure 6: What is the right panel here? I do not see it explained in the text. I see that it is "Reciprocity", but what do the percentages mean?

Page 15, Line 3-4: "sensitivity analysis with about 130 m, for the small profiles and 1300 m for the long one " This is confusing – please reword and check to be sure punctuation and word usage is accurate.

Figure 8: This is unnecessary as a stand-alone figure. The information here should be combined with Figure 3.

Page 17, Line 5: "stadiums" this is unusual usage of the word. Suggest replacing with a more common word.

Page 17, Line 10: How is the depth of investigation calculated?

Figure 9 and Figure 10: It seems that some masking is missing from the panels of this figure. Surely the Depth of Investigation could not be equal along the entire line length of all lines?

Page 20, line 12: "an excellent permeable channel for deep fluids conduct" – this is confusing as written, please reword.

Page 20, Line 12-14: This should be moved to the discussion.

Page 20, Line 24-27: References should be added to support this statement.

Page 21, Line 16-17: Please indicate on which ERT image this can be seen, and where on the image.

Page 21, Line 28: Is there any reference to support this supposed circulating mineral water?

Page 22, Line 3: "At at least one spot along our profile, the HMF, these fluids can propagate to the surface through the Tertiary sediments, but also at other sites expressions of fluid flow can be observed. " Please explain how this can be observed in the data measured for this experiment.

Page 22, Line 15-17: Suggest to support this statement with a reference.

Figure 11 (and reference to Figure 10): It is well known that inversions can result in over- or under-estimations of physical properties across sharp boundaries. For example, on Figure 10, from 3000 – 5500m along the line, there is a change from resistive material to conductive z=0 to z = 200 m. Here in figure 11, this is interpreted as "lower clay and sand" in a distinct unit – but how do you know this is not just an inversion smoothing artifact?

Page 23, Line 3: Figure 11 should be explained in the discussion, not conclusion.

The conclusion section contains "summary" content and "discussion" - please rewrite this to focus on only concluding remarks.

―――――――――――――――――――――

---

## Referee Comment (RC2) · Anonymous Referee #2 · 31 Mar 2019

General comments:

The paper describes an application of electrical resistivity tomography to image structural features in the Cheb Basin, targeted to identify fluid-related structures. Its application of a large-scale survey in itself is quite novel, and the results agree well with borehole logs. Although the authors state that the main target is to image fluid-related structures, the paper really describes a more structural characterization of the Cheb basin by integrating large-scale resistivity, gravity, borehole, and geological information. While the geophysical data agrees well with the borehole logs, the contribution

of the geophysics to the development of the geological model remains unclear, as the added benefit of the geophysical investigation is not clear.

What also remains somewhat unclear is why the authors actually choose to use ERT? There are other, i.e. EM methods, that may be more suited for this kind of deep investigation of resistivity structures.

More generally, the logic of the paper should be improved. This is clear when considering the Figure ordering, referencing, and placing, where, e.g., Fig. 2 is referenced before Fig. 3, and Fig. 1 is about 5 pages after it has been referenced first.

Specific comments:

One of the reasons for the limited benefit of the geophysics may perhaps be the large regularization factor that was applied to the resistivity inversion. This, in turn, led to a rather smooth resistivity model, which agrees well with the already existing borehole logs, but other than hinting to a basaltic intrusion, adds only limited new information. Perhaps more or an adapted data filtering may be required to help to achieve an acceptable $Chi^2$, while having a lower regularization factor. The authors are not providing any information on the sensitivity distribution or DOI index (e.g. Oldenburg & Li, 1999), which would allow to judge the reliability of the resistivity models particularly in depth. Providing a more thorough analysis and description of the resistivity models may help to improve the value of the geophysical data to the geological model development.

Regarding the title, the authors state that they are investigating "fluid-related structures". As resistivity depends on several factors, this relation from the resistivity model to fluids remains questionable. Especially given the geology of the study site, where the clay-rich Cyprus formation may well show the same response as a hydro-thermally altered rock formation.

Technical comments:

P1, Line 6: This is somewhat confusing. Why do you require a deep drilling program to

study near-surface structures? Near-surface is perhaps a subjective phrase depending on the audience.

P2, Lines 11-12: This sentence interrupts the flow here, as in the following sentence you provide more detail on the activities described before. Also, it might be worth adding what the open questions are.

P3, Lines 6-7: Why is a dipole-dipole array a "special investigation strategy"? I would describe this as a standard ERT array.

P3, Lines 28-30: You should reference to Figure 3 here.

Section "Geology and geodynamic activity": This section is very detailed and can be shortened by focusing on the main processes that are causing the swarms and CO2 release.

P6, Lines 21 – 26: Since you refer to the results here, it would be good to also show them.

Figure 1: Dashed line (i.e., country border) should be included in the legend

P8, Line 8: You refer to Fig. 3 before Fig. 2. Please revise your order of figures, which doesn't seem very logical at the moment.

P8, Line 13: These are good examples, but since you are referring to novel techniques, this list isn't exhaustive.

P8, Line 20: Although the practical reason is obvious to me, i.e. electrodes of the injection dipole need to be connected to each other, the theoretical reasoning is not as other arrays may achieve deeper penetration or higher resolution.

P12, Line 7: Do you mean that you assume that the signal is not distorted, hence has a very high signal-to-noise ratio?

P12, Line 10: What is alpha?

P14, Lines 1-2: Please clarify, what do you mean by this? Do you mean that you distinguished bad data points by their corresponding reciprocal error? Or do you mean that most of the bad data points have a good quality reciprocal measurement?

P14, Line 8: How do you deal with measurements that don't have a reciprocal measurement? Are you estimating an error model from the reciprocal data or are you assigning measurement errors otherwise?

P15, Line 9-10: If no error estimate is available I would suggest not including error weights in your inversion. Adding the BERT default is likely not your actual error model, and will have an impact on your inversion result.

P15, Line 14: This is quite a large regularization parameter and will likely result in very smooth models. Did smaller values result in much higher misfits? Did you do a L-curve analysis?

P15, Line 19: This is only true if the outlier also has a high error, otherwise the high regularization factor is likely causing the smooth response.

P17, Line 10: This would be more obvious if you add the sensitivity distribution, e.g. as shading.

P17, Line 11: Although most of them are not exactly on the line, could you add simplified logs to Fig. 9?

Figure 10: As for Figure 9, I suggest adding either the sensitivity distribution or calculating a depth-of-investigation index to quantify a "reliable" depth of your ERT models.

P 20, Line 11: Since you are referring to the gradient here, it might be worth plotting it as well.

Figure 11: Other than the possible basaltic intrusion, what is the contribution of the ERT and gravity measurements to this model? Especially the PPZ doesn't seem to show an expression in the data.

P23, Lines 11-13: I don't think this conclusion is obvious from your data. Why couldn't it be related to a thickening of the Cyprus formation?

References: Oldenburg, D. W., & Li, Y. (1999). Estimating depth of investigation in dc resistivity and IP surveys. GEOPHYSICS, 64(2), 403–416. https://doi.org/10.1190/1.1444545

---

## Author Comment (AC1) · 27 May 2019

Dear referee,

please accept the attached document as our reply to your comments. Our reply contains every remark/suggestion/question from your original review, our reply below and how we changed the text accordingly with quotations or the reworked figures. We appreciate that you have taken the time to read our manuscript and we have taken your remarks and comments seriously. We implemented your suggestions and have replied

to each and every comment. Thank you for improving our manuscript which should be easier to access after the rework.

Sincerely Tobias Nickschick and colleagues

Please also note the supplement to this comment:
https://www.solid-earth-discuss.net/se-2019-38/se-2019-38-AC1-supplement.pdf

**Supplement:**

Dear referee,

Thank you for reviewing our manuscript. We appreciate your comments and suggestions and have stated our comments and changes in the text below every comment.

This is a review of "Large-scale electrical resistivity tomography in the Cheb Basin (Eger Rift) at an ICDP monitoring drill site to image fluid-related structures", by Nickschick et al. that aims to use geophysics to image fluid relevant structures in deep formations. This is an interesting application of a rarely-applied deep electrical imaging method and seems to be within the scope of the journal. The manuscript is written in acceptable English and the figures are well drafted, though several of the figures could be combined. The organization is adequate, but could be improved. Given that the claimed focus of the work is to elucidate fluid-related structures, I find that C1 there is relatively little treatment of this subject in the interpretation and discussion. Specific comments related to each of these general observations may be found below. I recommend that the manuscript be returned to the authors for revisions.

General comments: 1) strengthen the interpretation and discussion of fluids, or recast the purpose of the work towards structures (or whatever else seems most appropriate). 2) combine figures as noted 3) reorganize the text as noted, specifically focus on making the introduction flow better, ensuring that all content is in the appropriate section, and shortening the background section 4) given my comments below related to the complexities of interpreting ERT data due to convolved signals from porosity, chemistry, saturation, and clay, I suggest adding a focused section to the discussion (or interpretation) section clearly explaining how you tease apart these elements in your data.

We have focused more on the interpretational part and your suggestions about rearranging our text. More information about the local lithological and petrophysical properties are now provided and we stressed the relevant information. We reworked the figures and tried several combinations of your suggestions. Please bear in mind that we had to keep the figures large enough to be readable. Moreover, we had another critical evaluation of our text structure and argumentation chain and reworked it according to both referees' comments. Please refer to each specific comment below for more information.

Specific comments:

Introduction: the structure of the introduction is awkward, particularly because it immediately jumps into site description, without giving any big-picture setup or explanation.

Please understand that in fact the site is of utmost importance for using this method, this is why we start with the site and not the method. Understanding the situation foremost provides essential information about the "whys" and "hows". Stating the situation, the overall problem and this study's place in the overall context of the ICDP initiative - featuring several different (bio-)geoscientific projects – allows us to put our rather unconventional method and setup into the right light.

Page 1, L12: "series of open questions" Either state those questions here, or move this text to where the questions are stated.

We omitted the confusing sentence. The open question about the magmatic ascent and $CO_2$ degassing are not to be confused with the key questions we stated for this geoelectric survey.

Page 1, Line 20: Change "drills" to "drilling"

Done.

Page 2, Line 33-34 & Page 3, Line 1: While I certainly agree that ERT is sensitive to fluids as indicated here, this justification for the ERT method seems incomplete because the several earth properties that control resistivity can be difficult to tease apart to attribute. As indicated on P2L34, the measurement is sensitive to porosity, salinity, saturation, and clay content all at the same time, and therefore the only way to retrieve any one of these parameters is to know three others. Section 2.1 is very long and covers a wide variety of topics. Readability would be improved if this section was shortened and focused specifically on the topics most related to the manuscript.

We now include a table of the rare petrophysical parameters known from other studies (Dobeš et al. 1986), see comment below. Also, additional information (logging data) from a recent study (Bussert at al. 2017) about the topmost layer of weathered phyllitic basement in the fluid ascent zone is added. As mentioned, the complex interaction of porosity, salinity, saturation, and clay content is not trivial, and we are confident to make our statements more plausible. Please remember, that this experiment is about studying the subsurface resistivity distribution to find potential fluid pathways and/or fluid caused interactions with the rocks (geological situation). We also omitted several sentences that are not immediately important for our experiment, such as information about swarm earthquakes etc.

P6L32 – P7L1: Suggest breaking this into two sentences.

Done. We restructured this part and omitted the part about the Marianske Lazne Complex and the Tepla Barrandian, as this also fits the category "too much geologic information" (see previous comment).

P7L3: Suggest to add a reference to support this statement on low resistivity areas.

We added references that both describe the fluid-induced alteration to clay minerals in general and the inferred resistivity changes related for the target site. See also next comment.

P7L4&5: The topic of MT surveys was introduced back on Page 6: This text here seems repetitive, I suggest to reorganize or reword.

We agree that this lead to unnecessary confusion and thus reworked it. It now states:

"With the aim to reach deeper structures up to 5km, several magnetotelluric investigations in the western margin of the Bohemian Massif and along the 9HR seismic profile (Cerv et al., 1997, 2001; Pícha and Hudeková, 1997; Di Mauro et al., 1999) have been carried out since the 1990 resulting in very coarse conductivity models."

instead of a whole paragraph as before.

P8 L2: suggest to delete: "imaging a pathway from"

Done.

P8 L6-8: This text seems out of place. The authors have used this section to explain existing data, however this short paragraph indicates availability of data and explains their method for using it but does not explain the data. Could be rewritten to be more appropriate for this section, or moved to methods.

Previously, we did not present key facts for the experiment here. We have added the relevant information here in form of a short paragraph about the assumed petrophysical properties, based on former drill sample and log measurements from Dobeš et al. (1986) and recent log data from Bussert et al. (2017). The text now states:

"In addition to this geological constraint, we regarded the results from Dobeš et al. (1986): Their report contains valuable petrophysical information from previous studies about the different stratigraphic units in and below the Cheb Basin which we have summarized in Tab. 1. The phyllitic-granitic basement is characterized by low porosities of less than 5% compared to the sedimentary deposits on top, which feature porosities of 15-30%. Resistivity, however, may vary drastically, depending on heterogeneities within the sediments and whether fluids such as mineral waters or $CO_2$ are present or not and the report does not specifically state where the samples were taken from. For this area, Bussert et al. (2017) provides additional information. Not only do they mention the occurrence of highly mineralized water in the central part of the HMF, their geophysical log of the HJB-1 drill reveals resistivities of 5-10 Ωm for the sediments of the Cypris formation and 10-20 Ωm for the topmost part of the weathered phyllites. They are about one order of magnitude lower than the values presented in Dobeš et al. (1986) - stressing the importance of regarding the occurrence or absence of fluids even more."

Table 1. Petrological description of the stratigraphic layers of sediments and basements below the Cheb Basin, translated from Dobeš et al. (1986)

| Name of stratigraphic unit | rock type | Porosity [%] | electrical resistivity [Ωm] | |
|---|---|---|---|---|
| | | | minimum-maximum | average |
| Vildštein | gravel, sand, clay | 30.0 | 14-1600 | 350 |
| Cypris | clay, silt, carbonates | 14.5-21.5 | 50-1500 | - |
| Main Seam | lignite, sand, clay | 22 | 7-50 | 15 |
| Lower Sand & Argillaceous | gravel, sandy clay | - | 3-150 (depending on saturation) | 7.5 |
| Phylliitic basement | weathered phyllite | 3.2 | 75-140 | 110 |
| Phyllite basement | unweathered phyllite | 1.0 | 500-1800 | 890 |
| Granitic basement | granite | 5.0 | 65-650 (weathered); > 650 for unweathered | - |

P8-L10-12: As indicated above, the nature of ERT interpretations is that these several properties all affect the measurement together, and therefore it is difficult to point to any one contributor as the primary control on electrical properties. Large porosity could have the same effect as high conductivity fluid in small pores. Low saturation could have a similar affect to small porosity. I think it is inaccurate of the authors to say "ERT is qualified for the detection of fluid signatures" without carefully explaining this statement in the context of how each material fraction contributes to the measured electrical signals.

> We agree that our argumentation seemed a bit weak without presenting more specific information. To substantiate our point, arguments describing the available information (and limits) of certain parameters were added to the manuscript. The area here is rather specific and thus, our general statement that ERT is a tool to detect fluids in general is not well-written. As a source, we have the article of Dobes et al (1986) featuring petrophysical studies (density, porosity, resistivity). For example, they determined the phyllitic basement to feature porosities of less than 5%, much lower than the Tertiary sediments (20-30%). Including this kind of information supports the statements made in the manuscript (see also comment before). But it should be mentioned that this published data are not clearly connected with information about depths of samples or logs – differences to our situation might occur.
> Also, we have implemented information about the mineral water earlier in the article, which should help the reader to understand the geologic situation for this site as the fluid-rock interaction plays a significant role (see comment before). It was previously only mentioned in the interpretation, yet provides essential information for understanding the target area's complexity – especially considering the very low resistivity encountered here (see comment P8 L6-8 and Page 21, Line 28).

P8 L20-21: ". . .for practical and theoretical reasons, most suitable for large-scale ERT experiments. . ." Please explain why, related to both practical and theoretical reasons. This seems like an important element of this manuscript given that such large scale measurements are so uncommon. It is also counterintuitive since Dipole Dipole configurations are well known to have poor signal to noise in comparison with nested arrays, for example.

> We included more information about the reason for this particular setup. It is correct that these large profiles are quite uncommon, but we chose a dipole-dipole setting for mainly logistic reasons, as this is the setup with a permanent layout of separate voltage dipoles and a moving current dipole that requires a minimum of cables and thus field effort.

1) An ERT profile of almost 7 km, crossing several streets, a large factory, dirt roads and agriculturally used fields in a rural area provides quite a challenge. A dipole-dipole setup allows us to connect only neighbouring electrodes with cables for both voltage readings and current injections and still allows for proper signals after appropriate data processing. Using configurations where several hundreds of meters of cable have to be pulled through shoulder-high crops was simply impossible.

2) We expected to see subvertically oriented structural changes in form of faults and potential fluid paths, which are known to exist from previous studies, thus choosing a configuration that is more sensitive towards that.

3) As this large-scale setup has been used in multiple areas before (up to 23 km profile length), a certain familiarity with the whole procedure was given to guarantee a proper workflow. Special statistical signal processing methods (drift correction, selective stacking, cross correlation) of the time series of potential differences are applied to improve clearly the signal-to-noise ratio.

In the text you'll now find the paragraph:
"The data acquisition was performed using the dipole-dipole configuration (AB MN, with A and B being the current injection electrodes and M and N being the potential electrodes) which is, considering the cost-effect-relation for practical and theoretical reasons, most suitable for this large-scale ERT experiment. Transmitter and receiver units are physically separated on two lines reaching maximum dipole separations of 6.5 km (Fig. 1) while keeping the total length of required cables to a minimum as only neighbouring electrodes have to be connected. Considering crop growth in June in this rural area and traffic by agricultural farming machines in general, other arrays are not effective with large cable spreads of several kilometers. Furthermore, we expected vertically oriented features (faults, "fluid channels"), as seen in previous studies (Nickschick et al., 2015), supporting the choice of using a dipole-dipole setup and achieving good results in previous studies at different location with a similar setup (Flechsig et al., 2010; Pribnow et al., 2003; Schmidt-Hattenberger et al., 2013)."

Figure 2: I suggest either merging this with Figure 1 or Figure 3 to make a 2-panel figure, OR perhaps merging all three to make a single 3-panel figure.

We have tried several combinations of these three figures. All three figures are quite important: Figure 1 serves as the overall background for our introduction and the geologic situation (magmatic processes, existence of the main geologic features of granitic intrusion, phyllitic basement and Tertiary deposits). Figure 2 is major source of litho-stratigraphic information which allows our interpretation (in combination with petrophysical information) and absolutely necessary. Figure 3 provides the local information that is crucial for understanding our measurement procedure (gaps sue to roads, regional railway, villages), shows the drill locations and important features like the degassing area of the HMF and the two main tectonic features.

However, we rearranged these figured. We switched figures 2 and 3 to separate the regional location from previous results and then going back to the location with everything included that is important to the experiment.

Page 9, L2: "greater distances" suggest to replace this with actual distance numbers.

Done.

Figure 4 seems unnecessary and could be deleted.

Agreed. We removed this figure.

Page 11, Line 9: Please deleted "A number of"

We deleted this.

Page 12, L22: "Figure 6" Which panel of Figure 6 is being described here?

So far, only the left column (Fig. 6a) had been described, which is changed now in the text (see comment below):

Page 14, L2: "(White Columns)" what does this refer to? Which figure?

This refers to the white areas in Figure 6a. We clarified this:

Fortunately, the missing data (white areas in the lower left triangle) are mainly available through their reciprocal counterparts in the upper right triangle. Before, we had white as "zero" AND "no reciprocity available" due to the chosen color scale. This has been changed by using another color scale that represents small absolute reciprocal errors in grey and to distinguish them from missing data in white. New Figure 6b (now Figure 5b).

[Figure]

Page 14, L3-4: What figure does this refer to? I assume #6. "appears significantly smoother" Smoother than what? How do you know it is "significant"? If referring to Fig 6, left panel, then I disagree – if the authors intend to make this argument, then it should be supported by a quantified metric.

> Agreed. It is "visually" smoother with fewer single outliers and more connected ones (linked with bad coupling and thus high noise). This allows us to disregard a chain of voltages and then prefer the mirrored values.

> Reformulated the sentence: The upper right triangle (i.e. where the voltage is measured east of the current injection in the west) appears smoother and features fewer single outliers as a result of higher artificial noise in the west and better coupling conditions in the east while featuring more connected outliers linked with single dipoles (e.g. AB electrode pair 44-45, 56-57, 57-58).

Figure 6: What is the right panel here? I do not see it explained in the text. I see that it is "Reciprocity", but what do the percentages mean?

> Correct, Figure 6b was not explained in the text. This is now done and later Fig. 6b is explained accordingly:

> "In theory, every combination of current and voltage dipole is measured twice by taking into account the principle of reciprocity, which states that voltage and current can be interchanged. By comparing the apparent resistivity values for forward (AB dipole ahead of MN), $\rho^a_f$, with the backward (AB behind MN) values $\rho^a_b$ one can compute the relative reciprocity error

$$ r = \frac{\rho^a_f - \rho^a_b}{\rho^a_f + \rho^a_b} $$

> The reciprocal error is displayed in Fig. 6b. Wide areas appear grey, i.e. forward and backward data agree very well. For some data with short spacing (near the diagonal) the values deviate from zero due to different coupling. Furthermore, there are quite a few areas of significant deviations, where one needs to be removed. In general, reciprocal errors increase with increasing dipole separation and reflect the decreasing signal-to-noise ratio as a result of the strongly decaying signal strength."

Page 15, Line 3-4: "sensitivity analysis with about 130 m, for the small profiles and 1300 m for the long one " This is confusing – please reword and check to be sure punctuation and word usage is accurate.

> We apologize, the misplaced comma made the sentence illogical.

Figure 8: This is unnecessary as a stand-alone figure. The information here should be combined with Figure 3.

> We agree that Figure 8 was not well-placed. We were not capable of including the regional Bouguer gravity into Figure 3 due to an overload of information otherwise. Station locations are described in the text and thus we have removed the figure completely.

Page 17, Line 5: "stadiums" this is unusual usage of the word. Suggest replacing with a more common word.

We used "stages" instead which should fit better.

Page 17, Line 10: How is the depth of investigation calculated?

We follow an approach of cumulative sensitivity after Christiansen & Auken (2012). The maximum model depth is chosen at the depth where the total sensitivity meets a relative value of 90% (Günther 2004), implemented in BERT as the default value.

Figure 9 and Figure 10: It seems that some masking is missing from the panels of this figure. Surely the Depth of Investigation could not be equal along the entire line length of all lines?

We added an alpha shading based on the coverage for both the small and the large profiles (Figs. 9 and 10). Therefore we also had to choose a different (rainbow-type) colormap.

New Figure 9 (now Fig.7):

[Figure]

New Figure 10 base map (now Fig.8):

[Figure]

Page 20, line 12: "an excellent permeable channel for deep fluids conduct" – this is confusing as written, please reword.

"Excellent" is indeed a very strong word, we rephrased the sentence. Additionally, we included the link to studies, who also underline this statement in this area.

The text now states:

"Such tectonic/structural zones form permeable channels for the deep fluids conduct and have been mentioned before for this area Bankwitz et al. (2003b); Kämpf et al. (2013); Bräuer et al. (2008); Fischer et al. (2014, 2017)."

Page 20, Line 12-14: This should be moved to the discussion.

We have provided additional references. We do not interpret this based on our survey, we have merely linked the existing information from other studies and the existence of these faults to make the reader be able to follow our description of the gravity curve.

Page 20, Line 24-27: References should be added to support this statement.

We have added the reference to our presentation of the geologic transect as well as the relevant literature:

"Stratigraphic records mention the occurrence of phyllite at the base, yet it is described to be very heavily weathered/altered (Dobeš et al., 1986; Špicáková et al., 2000; Fiala and Vejnar, 2004; Bussert et al., 2017)"

Page 21, Line 16-17: Please indicate on which ERT image this can be seen, and where on the image.

This can be observed in our presentation of the small ERT, profile P2. We have also included the link in the revised text.

Page 21, Line 28: Is there any reference to support this supposed circulating mineral water?

Reliable information is scarce for this specific area. While on a regional scale, several spas exist in Karlovy Vary, Františkovy Lázně, Mariánské Lázně, Bad Brambach and Bad Elster and mineral and healing water is well-researched there, specific data is scarce for the area around our profile. The most reliable study is Bussert et al. (2017), that describes the HJB-1 drill in the center of the degassing. They describe water with an electrical conductivity of around 6800 µS cm$^{-1}$ with a chemical mixture of Karlovy Vary and Františkovy Lázně-type water. While drilling they found pressurized horizons which act a fluid barriers, but at tectonic faults, these can malfunction. Furthermore, our profile is very close to the Soos natural reserve (Fig 1) in which we can observe several different mineral springs close by.

We added this information about the springs and nature reserve at this point and extended this paragraph which now reads:
"One key aspect in the low resistivities we observe might be related to circulation and ascent of heavily mineralized water and CO2-rich fluids. Bussert et al. (2017) mention pumping tests at the HJB-1 drill site within the main degassing area around Hartoušov and, after drilling

through a caprock-like layer and hitting a supposed aquifer at 79-85 m, encountering subthermal mineral water with a high conductivity of around 6800 µS cm$^{-1}$ (about 1.5 Ωm). Especially the more porous sandy parts within the Tertiary deposits are aquiferous and penetrating them resulted in a sudden outburst of gaseous CO2 and water (Bussert et al. 2017). While especially the pelitic layers can be considered impenetrable to ground water, intense tectonic faulting is made responsible for the mixture of groundwater with deeper water-bearing formations along faults, joints and chasms and also with the aquiferous Lower Argillaceous-Sandy and Main Seam formations (Dobeš et al., 1986; Peterek et al., 2011; Bussert et al. 2017). This is stressed by geoelectric borehole logging in the HJB-1 drill at the HMF where throughout the Tertiary sediments resistivities of 5-10Ωm were measured and even within the topmost layers of the (weathered) basement (phyllite) resistivities did not exceed 20Ωm. Another, prominent example for the complexity of the hydrological situation is the close-by Soos Nature Reserve, which is just about 3 km to the NW of our survey profile (Fig. 2. Other mineral and ochre springs and mofettes are found within a few kilometers (Weinlich et al., 1998; Bräuer et al., 2005; Kämpf et al., 2013) and Karlovy Vary, Františkovy Lázne, Mariánské Lázne, Bad Brambach and Bad Elster are well-known for their spas and diverse mineral water sources."

Page 22, Line 3: "At at least one spot along our profile, the HMF, these fluids can propagate to the surface through the Tertiary sediments, but also at other sites expressions of fluid flow can be observed. " Please explain how this can be observed in the data measured for this experiment.

> This is not well-expressed from our side, we apologize. After rewriting this sentence and adding references, it should be clearer
>
> "Along our profile at the HMF, these fluids can propagate to the surface through the Tertiary sediments along the assumed course of the PPZ, but also at other sites expressions of fluid flow can be observed (Weinlich et al., 1998; Kämpf et al., 2013; Bräuer et al., 2014)."

Page 22, Line 15-17: Suggest to support this statement with a reference.

> The reference to this can found in the preceding sentence.

Figure 11 (and reference to Figure 10): It is well known that inversions can result in over- or under-estimations of physical properties across sharp boundaries. For example, on Figure 10, from 3000 – 5500m along the line, there is a change from resistive material to conductive z=0 to z = 200 m. Here in figure 11, this is interpreted as "lower clay and sand" in a distinct unit – but how do you know this is not just an inversion smoothing artifact?

> In this case, as in many others, we have the drill logs as a verification tool. The inversion was specifically done without constraints to cross-correlate "hard" evidence subsequently, which indeed worked very well. We have included the drill names in our presentation of the large-scale profile for better presentation purposes for the reader.

Page 23, Line 3: Figure 11 should be explained in the discussion, not conclusion. The conclusion section contains "summary" content and "discussion" - please rewrite this to focus on only concluding remarks.

We apologize for the layout error due to LaTeX trying to find a good spot for the figure. It should now be found in the interpretation chapter. For our last remarks, we removed the summary parts and limited it only to conclusions.

---

## Author Comment (AC2) · 27 May 2019

Dear referee,

please accept the supplemented document as our reply to your comments. The reply contains every remark/suggestion/question from your original review, our reply below and how we changed the text accordingly with quotations or the reworked figures. We appreciate that you have taken the time to read our manuscript and we have taken your remarks and comments seriously. We implemented your suggestions and have replied

to each and every comment. Thank you for improving our manuscript which should be easier to access after the rework.

Sincerely Tobias Nickschick and colleagues

Please also note the supplement to this comment:
https://www.solid-earth-discuss.net/se-2019-38/se-2019-38-AC2-supplement.pdf

**Supplement:**

Dear referee,

Thank you for reviewing our manuscript. We appreciate your comments and suggestions and have stated our comments and changes in the text below every comment.

General comments:

The paper describes an application of electrical resistivity tomography to image structural features in the Cheb Basin, targeted to identify fluid-related structures. Its application of a large-scale survey in itself is quite novel, and the results agree well with borehole logs. Although the authors state that the main target is to image fluid-related structures, the paper really describes a more structural characterization of the Cheb basin by integrating large-scale resistivity, gravity, borehole, and geological information. While the geophysical data agrees well with the borehole logs, the contribution of the geophysics to the development of the geological model remains unclear, as the added benefit of the geophysical investigation is not clear. What also remains somewhat unclear is why the authors actually choose to use ERT? There are other, i.e. EM methods, that may be more suited for this kind of deep investigation of resistivity structures.

More generally, the logic of the paper should be improved. This is clear when considering the Figure ordering, referencing, and placing, where, e.g., Fig. 2 is referenced before Fig. 3, and Fig. 1 is about 5 pages after it has been referenced first.

Finding potential fluid- related structures, means having to characterize the area's structure, obviously. Having been successful in previous studies using geoelectrical methods in the first hundred meter (Flechsig et al. 2008 and Nickschick et al. 2015), we used an uncommon, large-scale setup ( ~7km profile). Of course, there is always the debate of which method to use. But, in general all EM methods including geoelectrics have as potential methods the same basic disadvantages.

We needed a method that is sensitive to fluid-induced effects at depths were borehole data does not exist (in general more than 200 meters). We focused on a depth scale of ~1000 m with a spatial resolution of 50-100 m. On a more regional scale, MT measurements (with a site spacing of 2 km) had been done (Munoz et al. 2018). However, these studies showed the problems for this area: High industrial/anthropogenic noise by lignite mining, agricultural usage with heavy machines, electrified railroads etc. Having to use farmland during crop growth season for a big part also does not allow using large coils amidst the fields and crossing the roads. In case of our setup and strategy, the specialty is an adapted statistical data processing which improves the signal-noise relation also for dipole-dipole measurements.

Specific comments:

One of the reasons for the limited benefit of the geophysics may perhaps be the large regularization factor that was applied to the resistivity inversion. This, in turn, led to a rather smooth resistivity model,

which agrees well with the already existing borehole logs, but other than hinting to a basaltic intrusion, adds only limited new information. Perhaps more or an adapted data filtering may be required to help to achieve an acceptable Chiˆ2, while having a lower regularization factor. The authors are not providing any information on the sensitivity distribution or DOI index (e.g. Oldenburg & Li, 1999), which would allow to judge the reliability of the resistivity models particularly in depth. Providing a more thorough analysis and description of the resistivity models may help to improve the value of the geophysical data to the geological model development.

> The chi^2 is actually acceptable, i.e. the data can be fitted within noise as Figure 7 shows. There is some misfit, but only in the large dipole separations that have large errors and low weight anyway. We are very positive that one could not derive a significantly better model without disregarding lithological information. Improving settings would just mean finding other, equivalent, models. We have tried many different regularization approaches and strengths, but ended up showing the smoothest (easiest) model that fits the data according to Occams razor. Please note that the deepest boreholes end at z=100m a.s.l. and our image is much deeper (note the different aspect ratios of Figs. 2 and 10/11).
> We are now providing information on the sensitivity and DOI by alpha shading Figure 9 and 10 : see comments below.

Regarding the title, the authors state that they are investigating "fluid-related structures". As resistivity depends on several factors, this relation from the resistivity model to fluids remains questionable. Especially given the geology of the study site, where the clay-rich Cyprus formation may well show the same response as a hydro-thermally altered rock formation.

> This is absolutely correct. This is why we added the drill log data to be able to relate the resistivity distribution to actual. However, it should be mentioned that these published older data (Dobes et al. 1986) are not clearly connected with information about depths of samples or logs. Being able to translate resistivity into actual geologic information was a basic need for us and for the scientific community interested in that area. To limit the effects of parameters such as salinity, porosity, clay content, and fluid conductivity, we have added information and a table with available parameters:

> "In addition to this geological constraint, we regarded the results from Dobeš et al. (1986): Their report contains valuable petrophysical information from previous studies about the different stratigraphic units in and below the Cheb Basin which we have summarized in Tab. 1. The phyllitic-granitic basement is characterized by low porosities of less than 5% compared to the sedimentary deposits on top, which feature porosities of 15-30%. Resistivity, however, may vary drastically, depending on heterogeneities within the sediments and whether fluids such as mineral waters or $CO_2$ are present or not. For this area, Bussert et al. (2017) provides additional information. Not only do they mention the occurrence of highly mineralized water in the central part of the HMF, their geophysical log of the HJB-1 drill reveals resistivities of 5-10 Ωm for the sediments of the Cypris formation and 10-20 Ωm for the topmost part of the weathered phyllites. They are about one order of magnitude lower than the values presented in Dobeš et al. (1986) - stressing the importance of regarding the occurrence or absence of fluids even more."

Table 1. Petrological description of the stratigraphic layers of sediments and basements below the Cheb Basin, translated from Dobeš et al. (1986)

| Name of stratigraphic unit | rock type | Porosity [%] | electrical resistivity [Ωm] | |
|---|---|---|---|---|
| | | | minimum-maximum | average |
| Vildštein | gravel, sand, clay | 30.0 | 14-1600 | 350 |
| Cypris | clay, silt, carbonates | 14.5-21.5 | 50-1500 | - |
| Main Seam | lignite, sand, clay | 22 | 7-50 | 15 |
| Lower Sand & Argillaceous | gravel, sandy clay | - | 3-150 (depending on saturation) | 7.5 |
| Phylliitic basement | weathered phyllite | 3.2 | 75-140 | 110 |
| Phyllite basement | unweathered phyllite | 1.0 | 500-1800 | 890 |
| Granitic basement | granite | 5.0 | 65-650 (weathered); > 650 for unweathered | - |

Technical comments:

P1, Line 6: This is somewhat confusing. Why do you require a deep drilling program to study near-surface structures? Near-surface is perhaps a subjective phrase depending on the audience.

Now we are confused. The sentence stated that the ICDP project "Drilling the Eger Rift" focuses on the possible connection between fluids (especially the ascending $CO_2$ of mantle origin) and the swarm earthquakes. Within this ICDP project there are several projects that explore(d) the area and 5 drill holes up to 400 m. We have changed the "near-surface" part however, as you suggested.

P2, Lines 11-12: This sentence interrupts the flow here, as in the following sentence you provide more detail on the activities described before. Also, it might be worth adding what the open questions are.

This sentence seems to have caused several issues, we have removed it to avoid confusion with our key questions.

P3, Lines 6-7: Why is a dipole-dipole array a "special investigation strategy"? I would describe this as a standard ERT array.

Again, we have not expressed this very well. We meant special as "specific" not as extraordinary. The basic setup is a dipole-dipole array, but the measurement strategy is different to common measurements. We use a permanently placed array of single dipoles for the voltage registrations, a moving high power current source, and a subsequent data processing of the time series of voltage/current as input for data inversion.

P3, Lines 28-30: You should reference to Figure 3 here. Section "Geology and geodynamic activity": This section is very detailed and can be shortened by focusing on the main processes that are causing the swarms and CO2 release.

We are shortening this in the revised version. We thought it would help the reader to understand the multi-scale effect of the fluids/CO2, but that both referees prefer a shorter paragraph and thus we have shortened it.

P6, Lines 21 – 26: Since you refer to the results here, it would be good to also show them.

We would kindly ask to look at the references provided. Repeating existing data from other studies would not be appropriate.

P8, Line 8: You refer to Fig. 3 before Fig. 2. Please revise your order of figures, which doesn't seem very logical at the moment.

We have reworked the figures. We are sorry for the order of the figures as this seems to be caused by a LaTeX error and floating figures, we apologize and fixed this.

P8, Line 13: These are good examples, but since you are referring to novel techniques, this list isn't exhaustive.

We deleted the word modern and added an "e.g." to make clear that this list is not exhaustive.

P8, Line 20: Although the practical reason is obvious to me, i.e. electrodes of the injection dipole need to be connected to each other, the theoretical reasoning is not as other arrays may achieve deeper penetration or higher resolution.

We agree, we have worded this poorly. We now added information that shows the practical reasons (agricultural usage, roads, total length of cables needed) but from the "theoretical" perspective we expected vertically oriented structures (faults, vertical fluid channels) and needed a high sensitivity towards that. Having to inject several Amperes of current over several kilometers would also be impractical in this noisy area with factories, streets and villages.

In the text you'll now find the paragraph:

"The data acquisition was performed using the dipole-dipole configuration (AB MN, with A and B being the current injection electrodes and M and N being the potential electrodes) which is, considering the cost-effect-relation for practical and theoretical reasons, most suitable for this large-scale ERT experiment. Transmitter and receiver units are physically separated on two lines reaching maximum dipole separations of 6.5 km (Fig. 1) while keeping the total length of required cables to a minimum as only neighbouring electrodes have to be connected. Considering crop growth in June in this rural area and traffic by agricultural farming machines in general, other arrays are not effective with large cable spreads of several kilometers. Furthermore, we expected vertically oriented features (faults, "fluid channels"), as seen in previous studies Nickschick et al. (2015), supporting the choice of using a dipole-dipole setup and achieving good results in previous studies at different location with a similar setup (Flechsig et al., 2010; Pribnow et al., 2003; Schmidt-Hattenberger et al., 2013)."

P12, Line 7: Do you mean that you assume that the signal is not distorted, hence has a very high signal-to-noise ratio?

Again, we have worded this poorly. Yes, we meant exactly that and have already fixed this.

"This provides correct results in case of a symmetric signal with an identical positive and negative amplitude, which is given in this case by controlling the source and assuming that the signal is not distorted by having a very high signal-to-noise ratio."

P12, Line 10: What is alpha?

Alpha is the rejection rate of samples after stacking, as is stated.

P14, Lines 1-2: Please clarify, what do you mean by this? Do you mean that you distinguished bad data points by their corresponding reciprocal error? Or do you mean that most of the bad data points have a good quality reciprocal measurement?

We meant that we do not have all data as reciprocal pair because certain current injections were not possible (vertical white columns) or certain voltage data could not be successfully retrieved. However, most of the missing data are available at least by one of the AB-MN or MN-AB combinations. See also more extensive reply to the other referee.

P14, Line 8: How do you deal with measurements that don't have a reciprocal measurement? Are you estimating an error model from the reciprocal data or are you assigning measurement errors otherwise?

As written, we estimated a percentage error of 5% and a voltage error of 2μV from the reciprocals. If a reciprocal pair is present, we took the current-weighted average. If only one was available, we took that value. The reciprocity analysis is an additional quality check compared to "traditional" surveys where only one measurement is carried out, we have in >70% of all dipole pairings another value to compare to decrease possible outliers or missing values.

P15, Line 9-10: If no error estimate is available I would suggest not including error weights in your inversion. Adding the BERT default is likely not your actual error model, and will have an impact on your inversion result.

Even an imperfect error model is better than no error model (i.e. assuming all data have equal quality independent on the voltage), since it is clear that measurements with large voltages (and low geometric factors) are more reliable than low-voltage measurements (with high geometric factors). This routine has been widely accepted in ERT. See also comment to the other reviewer and the new text about the background of the reciprocity and the interpretation of Figure 6b.

… "Therefore it can be used as a measure of data consistence and also to derive error models (Udphuay et al. 2011), however only if a statistically large number of data is available."

P15, Line 14: This is quite a large regularization parameter and will likely result in very smooth models. Did smaller values result in much higher misfits? Did you do a L-curve analysis?

A L-curve analysis is not quite easy as the appearance of the L depends on the scaling and the range of the lambda values. We basically chose the lambda value high enough so that we could avoid artifacts (conservative approach or Occam's razor). A further reduction of lambda decreased the error only slightly and lead to more unrealistic structures in the model that was not helping the interpretation. We did a large number of different parameters with very similar results. Again, our model fits the lithological data very well for the first 300-400 meters.

P15, Line 19: This is only true if the outlier also has a high error, otherwise the high regularization factor is likely causing the smooth response.

The parts of the pseudosection that could not be fitted well are in areas of large dipole separation and thus high geometric factors and error levels (up to 20%, see above). Also, the model is in agreement from what can be derived from drill logs.

P17, Line 10: This would be more obvious if you add the sensitivity distribution, e.g. as shading.

We added an alpha shading based on the coverage for both the small and the large profiles (Figs. 9 and 10). Therefore, we also had to choose a different (rainbow-type) color map.
New Figure 10 (now Figure 8) base map:

[Figure]

P17, Line 11: Although most of them are not exactly on the line, could you add simplified logs to Fig. 9?

We did that as well. New Figure 9 (now Figure 7):

[Figure]

Figure 10: As for Figure 9, I suggest adding either the sensitivity distribution or calculating a depth-of-investigation index to quantify a "reliable" depth of your ERT models.

We did it (see comment and new Figure 10 (now Figure 8) above).

P 20, Line 11: Since you are referring to the gradient here, it might be worth plotting it as well.

As an exception we have decided to not include the horizontal gravity gradient here. Another plot for the gradient would overload the whole figure with additional, unnecessary information and the gradient could also be derived from the primary gravity curve, please take this as not to overwhelm the reader.

Figure 11: Other than the possible basaltic intrusion, what is the contribution of the ERT and gravity measurements to this model? Especially the PPZ doesn't seem to show an expression in the data.

We have decided to rework Figure 11. As you say, the impact of our survey is not visible at first glance so we changed that. We now have used the stratigraphic model but implemented the observed resistivity to show that even within the same lithologic unit it can change significantly which is new, especially for anything below 200-300 meters in the (unexpected deeply weathered/altered) basement. Having now the stratigraphy (colors) linked with the new resistivity distribution leads to our interpretation and that should be easier to grasp with the reworked figure.

New Figure 11:

[Figure]

P23, Lines 11-13: I don't think this conclusion is obvious from your data. Why couldn't it be related to a thickening of the Cyprus formation?

If this were the case, we would not observe a resistivity shift in the eastern part. It is very likely that here the Cypris formation is "dryer" than in the western half, which we have also underlined by adding references.

---

## Author Response (AR1)

**Large-scale electrical resistivity tomography in the Cheb Basin (Eger Rift) at an ICDP monitoring drill site to image fluid-related structures**

Tobias Nickschick1, Christina Flechsig1, Jan Mrlina3, Frank Oppermann2, Felix Löbig1, and Thomas Günther2

1Institute for Geophysics and Geology, Leipzig University, Talstrasse 35, 04103 Leipzig, Germany
 2Leibniz Institute for Applied Geophysics, Stilleweg 2, 30655 Hannover, Germany
 3Institute of Geophysics CAS, Boční II 1401, 141 31 Praha 4, Czech Republic

Correspondence: Tobias Nickschick (tobias.nickschick@uni-leipzig.de)

**Abstract.**

[revised manuscript text omitted]

---

## Author Response (AR2)

We are responding to the reviewer comments (black) in blue color and provide new or changed text in green color. Even though we do not fully agree on some of the points, all comments were considered and lead to changes in the text, which further improved the manuscript.

**2nd reply to referee#1**

Thank you for the critical comments that helped us identifying weaknesses in the manuscript. Even though we do not agree on some points, we now accompanied text changes for all points and hope that these will meet your expectations.

Some important clarifications, revisions, and improvements have been made in this version of the manuscript. However, in many cases, it was apparent that "changes" made in response to referee comments did not actually address the point made by the referee. In other words, changes were made, but they were irrelevant or partially relevant to the referee comments. In other cases, referee suggestions appeared to be outright declined. The following is a list of the most offensive examples of comments that were not duly considered and a short explanation of how the response was insufficient for each:

With all due respect, there was only one suggestion that we did not accept immediately – the one about combining figures. As we have stated in our very first comment, it seemed impractical to combine these figures into one as they didn't immediately fit together in view of later layout. We also tried to avoid single images with way too much information, but we have changed this now. This was not intended to "offend" you - it was merely a suggestion from our side. You will be happy to hear that we have followed your initial suggestion and merged Figure 1 as inset into Figure 2.

1) Original Referee Comment: "Introduction: the structure of the introduction is awkward, particularly because it immediately jumps into site description…"

Why the response is insufficient: No action was apparently taken in response to this comment. Rather, the author attempts to justify a poorly organized introduction by saying the site is the most important thing. This argument is flawed – a well organized introduction will not detract from the importance of various study components. A well organized introduction will certainly improve readability and comprehension by readers. Placing something "first" in the text has no effect of making it appear "more important."

Your statement was "the structure of the introduction is awkward, particularly because it immediately jumps into site description, without giving any big-picture setup or explanation".

In our initial reply, we tried to reason why we start with the area – as it is common in articles about the Cheb Basin/Eger Rift due to the unique setting. Indeed the Cheb Basin is the big picture you speak of – the interaction of tectonic and magmatic processes – and the approach is how and what we can do gain more insight into the topic of magmatic fluids.

As this seems to have caused confusion and you do not like our approach, we have now added a passage in the very beginning that should help the reader dive into the matter of subject:

"Over the last decades, methods that study the electric resistivity of the subsurface - such as magnetotellurics (e.g. Muñozet al. 2018 or Blecha et al. 2018) and Electrical Resistivity Tomography (e.g. Storz et al. 2000, Schütze and Flechsig 2002, Schmidt-Hattenberger et al. 2013) - have proven to be especially useful when fluids are involved, as they are used as efficient techniques for non-invasive imaging of subsurface structures. A lot of experiments that focus on carbon dioxide in particular have been carried out, mainly at carbon capture and storage sites, which are typically well explored and where the fluid injection system is controllable. However, when it comes to natural CO2 emanation sites, such as volcanically or magmatically active areas, this is often not the case: the fluid system can often be very complex and variable in space and time and hence requires special approaches. Furthermore, many methods that focus on fluids are often limited in their resolution and/or depth when it comes to studying those fluids, their migration and interactions with the host rock.

One major target site for this kind of studies is the western Eger Rift in Central Europe that has been a center of research for various mantle gas and fluid-related studies within the last 2 decades. It can be called a natural analogue to carbon capture and storage sites where methods used for the detection and monitoring of CO2 and fluids in general can be applied to great success (Schütze et al. 2012)".

2) Original Referee Comment: "Page 2, Line 33-34 & Page 3, Line 1: While I certainly agree that ERT is sensitive to fluids as indicated here, this justification for the ERT method seems incomplete because the several earth properties that control resistivity can be difficult to tease apart to attribute."

Why the response is insufficient: The response indicates that a table is not included (not requested in the review comment) and logging data has been added (also not requested). The remainder of the response is primarily statements unsupported by evidence. An explanation of how the various material properties can be determined from resistivity as requested was not attempted.

We indeed _have_ included a table, as is stated in the reply letter. Despite not being requested, we have added both the table and the logging information as hard evidence which are always more valuable than speculations or assumptions assumptions (as a response to various comments of both reviewers). You have not asked for how our experiment can be used to derive porosity, salinity, saturation, and clay content - that was also not the point of the experiment. You have stated that "this justification for the ERT method seems incomplete because the several earth properties that control resistivity can be difficult to tease apart to attribute" – which was not the point of the experiment to begin with.

We have included information about the porosity and salinity, as well as clay content from several available sources, which you now call "not requested".

We also are unsure what you mean by "the remainder of the response is primarily statements unsupported by evidence", as these are published studies from other people which were taken into account. These are not assumptions, these are findings from other researchers in this particular area.

Just because we have no control of the exact value for each of those parameters, one cannot question the method itself, especially when they are expected to vary throughout the area.

Otherwise, no field method would be ever applicable and deriving exact values for each parameter is impossible, for any method. For a more qualitative discussion, see next comment.

3) Original Referee Comment: "P8-L10-12: As indicated above, the nature of ERT interpretations is that these several properties all affect the measurement together…"

Why the response is insufficient: This is essentially the same comment as in #2 above, and the author response is essentially the same. While saturation, porosity, fluid conductivity, and clay content all contribute to the total observed resistivity, the author only partially addresses porosity. This is central to the interpretation of the manuscript because presumably the gas-phase they a looking for will have an impact on saturation, and pore water conductivity is unknown – these properties would certainly effect the total resistivity, however there is no adequate discussion of the point at all in the revision.

We have used as much information as possible. Because there is no information available for depths > 300m we can only speculate about the parameters. Hence, the text now states (in the discussion):

"Clear limitations of this experiment occur at depths where stratigraphic records do not exist and thus do not provide essential information, but several assumptions can still be made about the relevant parameters:

1) Clay content: Basically the whole basin and basement is dominated by clay minerals: The Vildstejn formation has a very high, but varying, clay content, the Cypris basin is made of mudstone and the basement consist of phyllite/claystone/mudstone, which are weathered and re-mineralized as clay minerals.

2) Saturation: the ground water level overall is very high, ranging from about 5m in the center to a few tens of meters at the eastern and western flank, otherwise the clay-dominated rocks can be considered saturated and considering our resolution of 100 m, this transition from unsaturated to saturated state is barely noticeable in the large-scale profile. The separation of free $CO_2$ from its solved state in water is interpreted at about 50 m, according to findings from Nickschick et al. (2015), but heavily depends on pressure.

3) Porosity: It is impossible to make exact statements about the porosity at a depth >300 m due to the complex interaction of lithostatic pressure, hydrostatic pressure, pore size change by mineral alteration, mobilisation and precipitation from the aggressive fluids and general occurrence of joints and chasms by the earthquake activity and the p-T conditions in this geodynamic setting. For none of those, information can be found for the lower two thirds of the model.

4) Fluid conductivity: This is highly variable. We have one single measured conductivity from Bussert et al. (2017) for this specific area for the mineral water of 6800 µS cm-1. However, we have to consider ground water, different mineral water springs, $CO_2$-bearing water (dissolved $CO_2$ as well as a high fraction of solved ions) free gas near the surface, supercritical $CO_2$ at some hundreds of meters, which also changes depending on the location and depth and their aquifers.

Studying these in the future will be interesting when additional data might be available from the ICDP drilling or seismic exploration. Until then, the high variability of each parameter can only be estimated very roughly with this large-scale experiment and its resolution."

Overall this is a highly complex situation with very little information about the requested parameters, especially at depths where drills do not provide any information. Be reminded that this is a field experiment in a very heterogeneous environment. This is a site survey before a major drilling.

We have also not specifically been looking for a gas phase. Due to the complexity, it is impossible to attribute the different states of $CO_2$ (dissolved, free, supercritical) in a tectonically and magmatically active area, where nothing is known about the p-T conditions, admixture of water-rock suspensions, the fluid feeding system from the magmatic intrusion etc.

4) Original Referee Comment: "Figure 2: I suggest either merging this with Figure 1 or Figure 3 to make a 2-panel figure, OR perhaps merging all three to make a single 3-panel figure."

Why the response is insufficient: The only change made was the re-order the figures. The justification given for not addressing the review comment is that "each figure is so important." To be completely clear: there is NO relationship between the "size or location of the figure" and "the importance of the figure." How could combining the three separate figures into one figure with three panels possibly detract from their importance? Combining figures that are similar or have similar context is an important tool to improve readability – it will not make the information seem less important.

We have adjusted to this by combining the first two figures so that the old Figure 1 is now an inset in the old Figure 2. This follows your initial request of "I suggest either merging this with Figure 1 or Figure 3 to make a 2-panel figure".

The geological transect still has to be a separate figure, as it is completely separate from the location and is part of the geologic situation.

[Figure]

Figure 1.Map of the measured large-scale ERT profile (6.5 km), small-scale 625-700 m long ERT profiles (P1, P2, P3), and existing drill holes (Czech Geological Survey) with lithological information. Red dotted line marks the location of the lithological transect in Fig. 2. The Počatky-Plesná zone (PPZ) and Mariánské Lázně fault zone (MLF) are drawn as the main tectonic features. HMF = Hartoušov mofette field.

Inset: Geological sketch map of the western Bohemia/Vogtland area and the Cheb Basin near the German-Czech border in Central Europe, modified from Flechsig et al. (2008); Dahm et al. (2013); Bussert et al. (2017).

4) Original Referee Comment: "Page 14, L3-4: What figure does this refer to? I assume #6. "appears significantly smoother" Smoother than what? How do you know it is "significant"? If referring to Fig 6, left panel, then I disagree – if the authors intend to make this argument, then it should be supported by a quantified metric. "

Why the response is insufficient: The review comment specifically requested a quantified metric to support the argument. This was outright ignored – no quantification is provide in the reformulated sentence.

Indeed, when arguing with the term "significant" one should prove this by a number. Therefore, we already had removed the word in the revised version, thus not ignoring the comment.

Considering the sensitivity distribution of ERT data, experts in the field have experience in data validity in terms of smoothness, but this is hard to describe. Therefore we are now describing the procedure that we have followed (but not described so far) in more detail:

In order to appraise the quality of the partially redundant sub-datasets, we used both data in preliminary inversion runs using default parameters. It turns out that the rms misfit of the upper-right triangle was explicitly lower (11%) compared to the lower-left triangle (27%) which shows more systematic structures in the misfit plot. Therefore, we decided to fill up missing data in the further by the latter.

5) Original Referee Comment: "Page 22, Line 3: "At at least one spot along our profile, the HMF, these fluids can propagate to the surface through the Tertiary sediments, but also at other sites expressions of fluid flow can be observed. " Please explain how this can be observed in the data measured for this experiment. "

Why the response is insufficient: The referee comments specifically requests that the author explains how this can be observed in the data measured from this experiment, however the response does nothing more than rephrase the sentence – there is still no indication of how this effect is directly interpreted from data or figures presented in the manuscript. Please explain how the data presented in this manuscript shows that fluids can propagate to the surface through tertiary sediments.

We have rephrased the sentence in the revision and it indeed does change the meaning:

"Along our profile at the HMF, these fluids can propagate to the surface through the Tertiary sediments along the assumed course of the PPZ, but also at other sites expressions of fluid flow can be observed (Weinlich et al., 1998; Kämpf et al., 2013; Bräuer et al., 2014)."

We observe fluid discharge in our experiment at the HMF directly in our experiment (small-scale ERTs, profile 2), and more in the Cheb basin, which can be seen in Weinlich et al., 1998; Kämpf et al., 2013; Bräuer et al., 2014, which are all special articles about fluids in the overall area. These fluid sites are a few kilometers away from our profile and thus cannot be shown in our data, hence the references to other, very well-known articles for that area. For the fluid ascent in the HMF specifically: it is a mofette field: a meadow filled with several visible (and hearable) $CO_2$ exhalation spots on top of our resistivity anomalies in profile 2 (small-scale ERTs)

Also, in the paragraph before, there is already the statement: "Other mineral and ochre springs and mofettes are found within a few kilometers (Weinlich et al. 1998, Bräuer et al. , 2005, Kämpf et al. 2013} and Karlovy Vary, Františkovy Lázně, Mariánské Lázně, Bad Brambach and Bad Elster are well-known for their spas and diverse mineral water sources."

Our large-scale experiment cannot show the fluid migration through the Tertiary sediments alone. It is not a continuous monitoring setup (and thus only a snapshot) and it is one of the conclusions we drew. But we know, from all the aforementioned references in the article that the CO2-dominated fluids at the surface that can be analyzed (geochemical composition and isotopes) are of mantle origin and thus have propagated through the sediments – mainly along

the confirmed faults (PPZ and MLFZ) and also via the numerous hidden ones (as described in the geological setting chapter). Also, bear in mind that our 100-m resolution setup did not allow the tracing of potential fluid pathways, unlike the small-scale ERTs (resolution of 5m compared to the large-scale one with 100m) and Nickschick et al. (2015 but these are limited to the near-surface (investigation depth of less than 100m).

Again, we never claimed to derive fluid motion from our experiment alone, but have used a multitude of different geoscientific studies to derive this, which is new for this area.

6) Original Referee Comment: "Page 23, Line 3: Figure 11 should be explained in the discussion, not conclusion. The conclusion section contains "summary" content and "discussion" - please rewrite this to focus on only concluding remarks."

Why the response is insufficient: The response explains a formatting error related to the placement of the figure: this is completely unrelated to the content of the comment which suggests ways to improve organization of the text.

This is not correct, we had indeed reworked the whole conclusion chapter, as it can be seen in the Markup changes file that we provided last time, and removed parts that belonged to the discussion. You were completely right about mixing summary and conclusion, but disregard that we've done as you asked. It is now half as long and now only contains what belongs in a "conclusion" chapter.

In addition, I remain unconvinced that this manuscript should be cast under the topic of "fluid pathways" as suggested by the Objective 2(p.6), and alluded to in the title. This point was made by both reviewers, but was ultimately not sufficiently addressed by the author. This is claimed to be accomplished in Figure 9, specifically through annotation with arrows that show interpreted fluid flow pathways. However there is no evidence in the geoelectrical image (Figure 8) of either 1) the PPZ/HMF nor 2) anything that could lead to the interpretation of the fluid flow arrows. Of course, the authors state in the abstract that "Distinct, narrow pathways for CO2 ascent are not observed with this kind of setup which hints at wide degassing structures over several kilometers within the crust instead." However this apparently contradicts what is indicated in the Fig 9 cartoon that shows CO2 movement along specific, narrow conduits.

The title of the first revision explicitly says "fluid-related structures", we already refrained from calling the originally used "pathways", as we do not observe them with the large-scale setup - that is correct. However, it was an objective of the study:" to image the possible fluid pathways and the feeding system of the degassing area: Which structures are linked to the migration of CO2? Do we recognize potential structures acting as a fluid trap?" – and it works for the near-surface, where the resolution is higher. We also observe structures limiting the fluid ascent.

With this setup, it was not possible to directly get a perfect standalone model based on ERT solely. Therefore we included many information from different geoscientific studies to improve the value.

For comments about the Figure, please see next comment.

Finally, the conceptual Figure 9 is essentially a redrawn version of the borehole cross section with resistivity values added for each lithology. Confusingly, the spatial variability of resistivity is noted in the labels (apparently following the associated comments in the Interpretation section), but each lithology is drawn in the same color so it is not possible to actually tell how the resistivity values are distributed as shown in the Figure 8 imaging. This leads back to another reviewer comment that was not adequately addressed: what value has the ERT image actually brought to the conceptual subsurface understanding? It seems that everything the authors want to interpret is taken from the borehole cross section (e.g., discrete $CO_2$ flow pathways that are not visible on the ERT image; overall subsurface structure and lithology that is better resolved by boreholes than from ERT; a 100 ohm m block is inferred on the conceptual model that was not observed in drilling, however it apparently has very little to do with fluid flow).

We can indeed tracke the influence of $CO_2$ on its surrounding in our experiment. This is, however, limited to the near surface (first 100m), where the small-scale ERTs with a resolution of 5 m are capable of tracing these emanations (profile 2 in the HMF in our small-scale ERTs). This was already done by Nickschick et al. (2015) in a comparative study of surface $CO_2$ measurements combined with gravity and ERTs.

Figure 9 (new 8) is a conceptual model, and yes, it includes the stratigraphic information. As a main difference, the stratigraphic information are from drills of 100-300 m depth, and our investigation goes far beyond that. The limited depth of the borehole compared to our investigation is visible in the presentation of our large-scale inversion model.

Our model extended to about 600 m, to be visually comparable to the initial transect. However, as the difference might not be striking enough, we extended the model to 1000m, as our investigation depth provides meaningful information about that depth. Furthermore, we refined the image in such a way that the resistivity image is implemented by different color saturations. We also redraw the pathways of $CO_2$ movement so that it becomes clear how the implications from the ERT experiment gain new insight. We hope that this model is now more striking and easier to understand, especially the different depth scales which seemed to have caused confusions.

[Figure]

Figure 8. Conceptual W-E model of the topmost 1000 m of survey area. Stratigraphic units are based on drill core information from Fig. 2 for the first 100-300 m. While the Cypris and Vildštein formations are characterized by higher resistivities in the eastern part, the basement features lower resistivities, which is attributed to CO2 ascent, rock alteration and highly mineralized water. Dashed lines are inferred faults. The first few meters of Quaternary coverage are not shown

Before that, most of the investigations only focused on the basin sediments, not on the Proterozoic basement. But here, the basement plays a key role in the whole fluid system: A package of 500-600 m of weathered/altered phyllite featuring very, very low resistivities of about 5 Ohm m, which has not ever been mentioned or discussed in any available source of literature in that area. The 100m block has indeed not been observed yet which is linked with the fact that no deep borehole exists in the vicinity.

Hence, we have reworked the figure again due to the referees' remarks and changed it in the revised version. It now has different colors again to visually represent the varying resistivities within the same stratigraphic unit.

As this didn't seem to be clear enough (for which we apologize): Our ERT experiment provides essential information about depths that drills did not reach. The text now also reads:

[revised manuscript text omitted]

---

## Author Response (AR3)

See below the response to the reviewer comments (black) in blue color and new or changed text in green color.

All changes are also visible in the version attached to this letter which features the markup changes from this version compared to the previous one.

**Reviewer #1 (Marceau Gresse)**

This article presents several 2-D ERT surveys performed in the Cheb Basin. One profile, 6.5 km long, allowed to reach a depth of investigation of 1 km, which remains unusual with D-C methods. Authors have done a careful work when processing/filtering the raw data which is appreciable.

This work brings interesting results that are quite consistent with boreholes data. In addition, the authors performed a gravity survey along a 2-D ERT profile and the anomaly seems reliable with the degassing area.

Authors are aware of ERT limitation, and globally made a good interpretation of the resistivity structure. However, there are 3 points I have raised about the inversion process where I request additional information. If these problems are cleared, I recommend to publish this work in SE.

Marceau Gresse

We appreciate your comments and thank you for reviewing this version of our manuscript. In this reply, we answer your questions and remarks to your content and are confident that this version has now improved the article.
* * *
Main points:

>> 1- During inversions, authors modified the classic vertical to horizontal smoothness factor. I wonder if the vertical conductive structures (so-called plume by authors) observed in Fig .6-2 are the results of the modified vertical to horizontal smoothness factor or not. What kind of structure is observed if a "classic smoothness factor" is used? What would be the RMS of a "regular inversion" VS the modified vertical smoothness factor? I request more explanations/results to justify the reliability of these vertical conductive structures.

The vertical smoothness factor zWeight (as explained by Coscia et al., 2011) represents, along with the global regularization strength lambda, our expectations to the subsurface in terms of smoothness and the predominance of layered structures. Experience shows that in layered environments (most common situation) an isotropic smoothness (zWeight=1) shifts the anomalies to great depth, which is why in BERT a default value of 0.3 is used. We tested several values for the small profiles where we had ground truth from boreholes (e.g. depth of Vildstejn formation in boreholes B-1 and HV-14 at the edges of profile P3, see Fig. 6) and decided for 0.2

as it agrees well with this data. Even lower values produced unnecessarily detailed structures, so we kept the value for all profiles. With other values, we can achieve the same RMS provided lambda is adjusted, so the data fit helps to determine the total amount of smoothness but not its anisotropy.

Previous studies (Flechsig et al. 2008, Nickschick et al. 2015, 2017) have shown that these vertical formations are stable features and only occur where strong $CO_2$ surface degassing is observed. In Nickschick et al. 2015, we used the standard smoothness factors and in this article here decided to use the modified version due to the better agreement with drill logs.

>> 2- Authors do not provide information about the RMS (or chi-square) evolution during inversions. It would be interesting to show the convergence of models in successive iterations, especially for the long 2-D ERT profile.

We added the RMS for each of the short profiles' last model (4.7%, 17.4% and 4.5%, respectively) and the iteration steps (3, 9 and 14, respectively) within the text in chapter 4.1 and within the caption of Figure 6.

For the long profile, we added the final RMS in the caption, but also modified a paragraph in chapter 4.2 which now states:

Figure 7 shows the inversion result of the long profile after 4 iterations (RMS evolution - 81%, 29.9%, 18.5%, 14.6% and 14.5% as the final RMS). On top, the lithology provided by the neighboring drills is plotted as colored boxes columns, indicating the limited depth that has been achieved by the drills.

>> 3- Page 15, line 30: I recommend to add additional information about the 1D sensitivity analysis used to determine the maximum depth of resistivity models.

We inserted the relevant citation and some text in the manuscript. It now states:

The maximum model depth has been determined by 1D sensitivity analysis as described by Günther (2004, section 3.3.2): the cumulative 1D sensitivity of all data for a homogeneous halfspace is summed up and the 90% threshold is considered as main subsurface contribution. This method yielded 130m for the small profiles and 1300m for the large profile.
* * *
Minor points:

>> Abstract

- Page 1, line 8: I would suggest to change "the presence of low-resistivity rock fraction… » by « The presence of conductive rock fractions…".

We have replaced this part of the sentence with your suggestion.

- Page 1, line 19: Authors explain that "gravity clearly shows the deepest part of the basin", but they do not present how (in the abstract). I think authors could add few words in this sentence to show the main result of this gravity survey. Example "A negative gravity anomaly of **** is observed along the long ERT profile and is explained/consistent by/with **** … showing the deepest part of the basin…"

> We agree and have now rephrased the sentence to also feature our most important discovery – the deep alteration/weathering within the basement. It now reads:

> A gravity anomaly of ca. -9 mGal marks the deepest part of the Cheb Basin where the ERT profile indicates a large accumulation of conductive rocks, indicating a very deep weathering or alteration of the phyllitic basement due to the ascent of magmatic fluids such as $CO_2$.

>> 1-Introduction

- Page 2, lines 5-7: I think authors could add few references about CO2 sequestration experiments.

> This is completely true, we have added some references for CO2 sequestration experiments which used ERT as their method of choice (Carrigan et al. 2009, Nakatsuka et al. 2010, Schmidt-Hattenberger et al. 2013, Bergmann et al. 2017).

- Page 2, lines 7-9: I fully agree with this sentence, but once again, I think the authors need some references to justify it. I suggest to cite some work combining electrical resistivity images and CO2 degassing surveys on volcanoes and/or geothermal areas (e.g. Gresse et al., 2017, Revil et al 2010).

> We also agree on this and have added your suggestion plus some more (Finizola et al. 2009, Pettinelli et al. 2010, Revil et al. 2008, Revil et al. 2011).

>> Figure

- Figure 1: Please add the coordinate system and its unit in this figure.

> We have added the longitude and latitude for the inset (if this is what you are referring to). Moreover, we added the reference to our coordinate (UTM 33N) into the caption.

[Figure]

**Figure 1**. Map of the measured large-scale ERT profile (6.5 km), small-scale 625-700 m long ERT profiles (P1, P2, P3), and existing drillholes (Czech Geological Survey) with lithological information. All coordinates are in UTM 33N. Red, dotted line marks the location of the lithological transect in Fig. 2. The Pocatky-Plesná zone (PPZ) and Mariánské Lázne fault zone (MLF) are drawn as the main tectonic features. HMF =Hartoušov mofette field. Inset: Geological sketch map of the western Bohemia/Vogtland area and the Cheb Basin near the German-Czech border in Central Europe, modified from Flechsig et al. (2008); Dahm et al. (2013); Bussert et al. (2017).

- Figure 6: It would be interesting to show the direction of each ERT profile. Information about faded colours (related to the depth of investigation) in the figure caption is also needed such as the final RMS of each model.

We are sorry that this has caused confusion. We have added directions for each profile on top as an indication for the orientation.

We used the coverage (cumulative sensitivity of the final model) for the alpha shading of the inversion results so that poorly covered model regions will not be interpreted. We refer to Ronczka et al. (2017) who substantiate the use of coverage as a simple approximation for the model resolution and by using synthetic and field ERT data. The relative RMS values are 4.7%, 17.4% and 4.5%, respectively.

[Figure]

The maximum resistivity value reported on the scale is 200 ohm m. However in the text, "4.1 -Small-scale ERT: Page 19, line 1. Authors wrote "Profile 3 (Fig. 6 bottom) reveals a 10-15 m thick layer with resistivities above 300 ohm m". Please correct the figure and/or the text.

> We have added this important part of information into the caption. We decided originally to have the color scale end at 200 Ohm as otherwise fine details in the other profiles cannot be shown adequately.

- Figure 7: It would be better to show the direction of each profile. Information about the faded colours in the figures caption is also needed such as the final RMS of each model.

> We have also added directions for each profile on top, provided the RMS value and noted the alpha-shading as we did in Figure 6. See also last comment for the shading.

[Figure]

- Figure 8: Authors wrote "m a. s. l." but they also use negative elevation values (below the sea level). Please change this unit.

We have re-edited the figure and used "elevation" instead of "m.a.s.l."

[Figure]

>> Other minor comments:

- Small Scale ERT, page 17, line 32 "…strong degassing at surface in previous studies…" should be replaced by "…strong degassing as reported at the surface in previous studies …"

    We apologize for this mistake, we have corrected this, of course.

- "Ground water level" should be written as "Groundwater level" or "Water table".

    We have found this error two more times and corrected in all three cases.

- Few "))" appear throughout the text.

    We found 5 occasions where 2 brackets occurred, we have fixed this and apologize for this oversight.

**Reply to Reviewer #1 - Anonymous**

This manuscript has been substantially revised and my previous concerns related to the earlier revisions have largely been resolved through targeted changes or clarified justifications. The authors may wish to consider making the following minor technical corrections in the final version of the manuscript.

We appreciate the kind words and are happy to see that our rework is satisfactory. We have of course fixed these mistakes and would like to express our gratitude again for being a helping us improving the manuscript.

Page 6, Line 1: Suggest to change "sensors" to "electrodes."

We replaces the word "sensors" here (and once more later on) with "electrodes".

Page 9, Line 5: It appears that a close-parentheses ")" might be missing after "Fig.2"

We have rephrased the sentence slightly and this is fixed now.

Page 23, Line 16: There is a space in the wrong place between "GPS" and "map," or perhaps no space is required.

Apologies for the misplaced blank space, this was now fixed.

Page 21, line 2: Space between "m" and "." at the end of the line could be removed.

The additional blank was removed.

[revised manuscript text omitted]